# The relationship of within-individual and between-individual variation in mental health with bodyweight: An exploratory longitudinal study

Julia Mueller[1]*, Amy L. Ahern[1], Rebecca A. Jones[1], Stephen J. Sharp[1], Alan Davies[2], Arabella Zuckerman[3], Benjamin I. Perry[4,5], Golam M. Khandaker[6,7], Emanuella De Lucia Rolfe[1], Nick J. Wareham[1‡], Kirsten L. Rennie[1‡]

1 MRC Epidemiology Unit, School of Clinical Medicine, University of Cambridge, Cambridge, United Kingdom, 2 Division of Informatics, Imaging & Data Sciences, School of Health Sciences, University of Manchester, Manchester, United Kingdom, 3 School of Clinical Medicine, University of Cambridge, Cambridge, United Kingdom, 4 Department of Psychiatry, University of Cambridge, Cambridge, United Kingdom, 5 Cambridgeshire and Peterborough NHS Foundation Trust, Cambridge, United Kingdom, 6 MRC Integrative Epidemiology Unit, Population Health Sciences, Bristol Medical School, University of Bristol, Bristol, United Kingdom, 7 NIHR Bristol Biomedical Research Centre, Bristol, United Kingdom

‡ NJW and KLR are joint last authors to this work.
* Julia.Mueller@mrc-epid.cam.ac.uk

**Data Availability Statement:** The dataset analysed during the current study is publicly available, and

## Abstract

### Background

Poor mental health is associated with obesity, but existing studies are either cross-sectional or have long time periods between measurements of mental health and weight. It is, therefore, unclear how small fluctuations in mental wellbeing within individuals predict bodyweight over short time periods, e.g. within the next month. Studying this could identify modifiable determinants of weight changes and highlight opportunities for early intervention.

### Methods

2,133 UK adults from a population-based cohort completed monthly mental health and weight measurements using a mobile app over a period of 6–9 months. We used random intercept regression models to examine longitudinal associations of depressive symptoms, anxiety symptoms and stress with subsequent weight. In sub-group analyses, we included interaction terms of mental health variables with baseline characteristics. Mental health variables were split into "between-individual" measurements (= the participant's median score across all timepoints) and "within-individual" measurements (at each timepoint, the difference between the participant's current score and their median).

### Results

Within-individual variation in depressive symptoms predicted subsequent weight (0.045kg per unit of depressive symptom severity, 95% CI 0.021–0.069). We found evidence of a moderation effect of baseline BMI on the association between within-individual fluctuation in

the host institution has an access policy (https://www.mrc-epid.cam.ac.uk/wp-content/uploads/2019/02/Data-Access-Sharing-Policy-v1-0_FINAL.pdf) so that interested parties can obtain the data for replication or other research purposes that are ethically approved. For ethical reasons as outlined in the host institution's access policy, we are unable to append an anonymized dataset, however data access is available upon reasonable request (datasharing@mrc-epid.cam.ac.uk).

**Funding:** The study was supported by the Medical Research Council (grant MC_UU_00006/1). GMK acknowledges funding support from the Wellcome Trust (grant no: 201486/Z/16/Z and 201486/B/16/Z), the UK Medical Research Council (grant no: MC_UU_00032/06; MR/W014416/1; and MR/S037675/1), and the UK National Institute of Health Research Bristol Biomedical Research Centre (grant no: NIHR 203315). The funder had no role in study design, data collection and analysis, decision to publish, or preparation of the manuscript.

**Competing interests:** JM is a trustee for the Association for the Study of Obesity (unpaid role). GMK receives royalties from Cambridge University Press (Editor of Textbook of Immunopsychiatry). ALA is the chief investigator on two publicly funded (MRC, NIHR) trials where the intervention is provided by WW (formerly Weight Watchers) at no cost outside the submitted work. ALA is on the Scientific Advisory Board for WW (payment to institution). The views expressed are those of the author(s) and not necessarily those of the NIHR, MRC or other funders. Huma Therapeutics Ltd. developed, validated and tested the app at no charge. This does not alter our adherence to PLOS ONE policies on sharing data and materials.

depressive symptoms and subsequent weight: The association was only apparent in those with overweight/obesity, and it was stronger in those with obesity than those with overweight (BMI<25kg/m$^2$: 0.011kg per unit of depressive symptom severity [95% CI -0.017 to 0.039]; BMI 25–29.9kg/m$^2$: 0.052kg per unit of depressive symptom severity [95%CI 0.010–0.094kg]; BMI≥30kg/m$^2$: 0.071kg per unit of depressive symptom severity [95%CI 0.013–0.129kg]). We found no evidence for other interactions, associations of stress and anxiety with weight, or for a reverse direction of association.

## Conclusion

In this exploratory study, individuals with overweight or obesity were more vulnerable to weight gain following higher-than-usual (for that individual) depressive symptoms than individuals with a BMI<25kg/m$^2$.

## Background

Whilst evidence indicates there is a bidirectional relationship between weight and mental health [1], the relationship is complex and remains poorly understood. Specifically, it is unclear how changes within individuals in mental health are associated with changes in bodyweight over time. This is important to understand as mental health may act as a potentially modifiable risk factor for subsequent weight gain, thus informing prevention and treatment approaches for overweight and obesity.

The majority of research to date on the relationship between mental health and weight has been limited by a reliance on cross-sectional studies [2–5], and extant prospective studies are limited by infrequent measurements of mental health and weight, with periods of typically one to several years between measurements [6–10]. It is, therefore, unclear how small within-individual fluctuations in mental wellbeing predict weight over shorter time periods, e.g. within the next month. Understanding how within-individual variation in mental health predicts subsequent weight can help us identify opportunities for early intervention to prevent weight gain. For example, if fluctuations in mental health predict weight change in the near future, we can use this to develop interventions that provide relevant support during "critical timepoints", when deteriorations in mental wellbeing occur, and before weight gain occurs. Interventions which adapt support over time to an individual's changing status (i.e. "just-in-time adaptive interventions") are increasingly being used in various domains (e.g. physical inactivity, alcohol use and smoking) and have potential to facilitate long-term behaviour change [11].

However, while previous research has indicated that mental health outcomes are associated with higher weight, BMI and weight-related outcomes such as food intake and reduced physical activity [6, 12–16], these studies make comparisons at the between-individual level, examining how average mental health is association with weight. For example, most studies examine how mental wellbeing at one timepoint predicts weight at a later timepoint. Thus, these studies treat mental health like a stable, trait-like construct and do not examine within-individual fluctuations in mental health over time and how these relate to changes in weight. Mental health is likely to change over time within-individuals; it involves a trait-like between-individual component which represents a person's general disposition, and a state-like within-individual component which represents a person's variation around their

general disposition. As such, between-individual comparisons which take only the trait-like component into account can mask important relationships [17] and limit our understanding of the relationship between mental health and bodyweight. Associations at the between-individual and the within-individual level can differ in both size and direction [18].

Thus, existing studies provide limited insights into whether intervening when fluctuations in mental wellbeing occur can prevent weight gain.

The lack of research on within-individual variation in mental health and weight is partly due to the paucity of datasets with frequent, repeated measurement data, but also because examination of change depends on the extent to which a given variable varies within an individual. When variation is low, it is difficult to assess how changes in one variable influence changes in another. During the COVID-19 pandemic, there were likely to be higher-than-usual fluctuations in weight and mental health over relatively short time periods [19–21]. Thus, the pandemic provides a unique opportunity to investigate the relationship between these variables. This can help provide a clearer picture of this association, which can be extrapolated to other situations of heightened psychological distress, thereby providing useful insights into the determinants of weight that are relevant beyond the context of the COVID-19 pandemic.

In the present exploratory study, we use data derived from a sub-study of the population-based and well-phenotyped Fenland study [22] which collected detailed information monthly using a mobile app over a period of 6–9 months during the COVID-19 pandemic. Using this information, we assessed the relationship between mental health (depressive symptoms, anxiety symptoms, stress) and body weight over an extended period. Importantly, we explored both how an individual's general mental health ("between-individual variation") may affect weight, but also how fluctuations in an individual's mental health compared to their own usual levels ("within-individual variation") may influence subsequent weight. This is the first longitudinal study on mental health and weight that specifically explores within-individual fluctuation in mental health, and the findings offer useful and timely insights for developing context-specific weight management interventions. Notably, we treat mental wellbeing as a symptom continuum, recognising that individuals may experience symptoms of mental illness without meeting diagnostic criteria [23].

We aimed to investigate:

i. whether within-individual variation in mental health is associated with weight at the next assessment timepoint (approx. 1 month later)

ii. whether between-individual differences in mental health are associated with weight

iii. whether the above associations vary by participant characteristics (age, sex, education, occupation, and baseline BMI)

iv. whether within-individual and between-individual differences in weight are associated with subsequent mental health (to investigate the reverse direction of association and help us better understand the direction of the association).

Objective iv) and the accompanying analyses were not included in the original Statistical Analysis Plan. However, on reviewing our findings regarding the effect of depressive symptoms on subsequent weight, and the moderating effect of baseline BMI, we deemed it important to explore the possibility of a reverse direction of association and assess whether weight predicts subsequent depressive symptoms.

## Methods

### Study design and procedure

Data used in the present study are derived from the Fenland COVID-19 app study, which is a sub-study of the main Fenland study [22]. Ethics approval was obtained from Southwest Cornwall and Plymouth Research Ethics committee. All participants gave written informed consent prior to participation.

**The main Fenland study.** The Fenland study is a population-based cohort study of 12,435 participants born between 1950 and 1975 which examines the interaction between genetic and environmental factors in determining risk of diabetes, obesity and related health conditions [22]. Participants were recruited from General Practice sampling frames in Cambridgeshire, United Kingdom (UK).

**The Fenland COVID-19 study.** The Fenland COVID-19 study was designed as an observational cohort with data collection over a period of 9 months during the COVID-19 pandemic. The main aim of this study was to determine the prevalence of previous infection with COVID-19 in this known population-based cohort [24]. Fenland cohort participants who were known to be still alive, had a valid phone number or email address and who had not withdrawn from the study were approached either by email, telephone, letter or text message (n = 11,469) and invited to take part from July 2020.

**The Fenland COVID-19 app study.** After giving written, informed e-consent to participate in the Fenland COVID-19 study, participants were sent further information on the app sub-study. This sub-study was designed in collaboration with Huma (https://huma.com). If participants agreed to take part in the app sub-study, they were sent a link to a second online consent form. Following informed, written e-consent, participants were sent an email with details to securely download and start using the app. They were asked to complete a variety of measures at different intervals. Participants consented into the Fenland COVID-19 study from 5th July 2020. Those in the app sub-study were able to download the app from 6th August 2020, and the study finished on 30th April 2021 (when the app also closed to further measurement entry). The number of monthly measurements participants could provide depended on the timepoint when they entered the study (e.g., a participant entering in August 2020 could add a total of 9 measurements up until April 2021, a participant entering in September could add 8, etc.). Participants received prompts on their phones to remind them to complete the monthly questionnaires. We chose monthly measurements since the mental health questionnaires enquired about the previous 4 weeks. For the present analysis, we use self-reported monthly data on mental health and weight that participants provided via the app, coupled with sociodemographic data collected during in-person visits completed as part of the main Fenland study. Authors did not have access to information that could identify individual participants during or after data collection.

### Study population

Inclusion criteria for the Fenland Study were i) date of birth between 01/01/1950 and 31/12/1975 and ii) registered with general practices in Cambridgeshire, UK. Further details are published elsewhere [25]. Participants in the app-study also needed access to a suitable smartphone (firmware iOS version 13.0 or above/Android 6.0 or above). For the present analysis, we excluded weight entries that indicated a biologically implausible BMI of <12 or >70kg/m$^2$ [26, 27] or biologically implausible weight loss between two timepoints as defined by Chen et al. [28]. We excluded participants who reported neither mental health nor weight data.

## Outcome

The outcome was body weight (kg). Participants were asked to report, monthly via the app, their weight in light clothing using the same weighing scales. Participants received specific instructions for measuring their weight accurately, including a detailed video.

## Exposures

Mental health was assessed monthly using three validated questionnaires (detailed below) via the app. For all mental health variables, we used the total score as a continuous measure of symptom severity.

**Depressive symptoms.** Depressive symptoms were measured using the Patient Health Questionnaire (PHQ-8), a validated self-report tool [29], which assesses depressive symptoms occurring in the past two weeks. Eight items are rated on a Likert-scale from 0 to 3, giving a total depression symptom score of 0–24. A higher symptom score represents greater depression severity. A score of ≥10 has a sensitivity of 88% and a specificity of 88% for major depression [30].

**Anxiety symptoms.** Anxiety symptoms were measured using the Generalised Anxiety Disorder questionnaire (GAD-7), a validated self-report tool [31]. Seven items are rated on a Likert-scale from 0 to 3, giving a total score of 0–21. A higher score represents greater anxiety severity. A score of ≥10 identifies generalised anxiety disorder (GAD) with sensitivity 89%, specificity 82% [31].

**Perceived stress.** Perceived stress was measured using the 10-item Perceived Stress Scale (PSS-10), a validated self-report tool [32]. The PSS assesses stress over the past month. Ten items are rated on a Likert-scale from 0 to 4, giving a total score of 0–40. A higher total score represents greater perceived stress. The measure is reliable and valid in a variety of settings [33].

## Confounders and effect modifiers

**Population characteristics.** We included age (continuous), sex (male/female), ethnicity (White/non-White), education (age when completed full-time education) and occupation (Traditional and modern professional and higher managerial/Lower managerial and intermediate occupations/Technical, semi-routine and routine occupations; categories are based on [34]) from the main Fenland study as covariates. We used height (cm) from the main Fenland study (measured in the clinic by trained staff) to compute baseline body-mass-index (BMI) using the first weight measurement provided by each participant.

**COVID-19 restriction level.** To account for confounding effects of restrictions due to the COVID-19 pandemic [35], we created an ordinal variable with three categories based on different levels of restrictions that were in place throughout the study timeframe, based on a timeline of UK coronavirus lockdowns provided by the Institute for Government [36]:

1. Minimal restrictions: from 1st June 2020 to 4th November 2020.

2. Some restrictions: from 5th November 2020 to 5th January 2021 and from 9th March to end of April 2021.

3. Strict restrictions: from 6th January to 8th March 2021

## Analysis

Analyses were pre-specified in a statistical analysis plan which was reviewed and agreed by the authors of the study prior to commencing analyses. We used R Version 4.1.0 for all analyses.

To describe baseline characteristics of the sample, we calculated means and standard deviations (SDs) for continuous variables, and the number and percentage of individuals within each category for categorical variables (using the number of non-missing values as the denominator).

Between-individual and within-individual effects of mental health domains were assessed on weight separately. We used the median of each person's monthly scores for the between-individual measurement and calculated the deviation of each monthly score from that person's median for the within-individual measurement. We used medians rather than means due to the skewed distribution of the mental health variables. To assess the effects of between-individual and within-individual variation in weight on mental health, we used the same approach, but using the mean instead of the median.

**Main models (mental health & weight).** We used random intercepts regression models to assess the relationship between mental health variables (depressive symptoms, anxiety symptoms, stress) and weight. Time (measurement time point), between-individual measurements and within-individual measurements of mental health variables, demographics (age, sex, education, occupation), baseline BMI, restriction level and season (month as dummy variables) were included as fixed effects. For the within-individual variables only, we added measurements of the previous assessment timepoint to the model (rather than of the same assessment timepoint as weight). This means that we assessed the association between within-individual variation in mental health with weight at the next assessment timepoint (i.e., one month later), rather than examining the association with weight at the same timepoint. We fitted separate models for each mental health domain. Participant unique number was included as a random effect. We used analogous random intercept models to assess the relationship of between-individual and within-individual variation in weight with subsequent mental health domains to explore the reverse direction of association. Additionally, to provide further context for the interpretation of the findings, we also fitted random intercept regression models for the association between change in mental health scores and change in weight (and the reverse), without splitting up the exposure variables into "within-individual" and "between-individual" variation. In these models, we regressed change in mental health scores from one timepoint to the next onto change in weight between the next timepoints (e.g., change in mental health score from month 1 to month 2 onto change in weight from month 2 to month 3). These analyses were not in the original statistical analysis plan, but were added on request during peer review to help the reader interpret the findings.

Random intercept models use all available data and assume that missing values are missing at random (MAR). To investigate this assumption, we used logistic regression to assess the association between participant baseline characteristics (age, sex, occupation, education, BMI, baseline mental health scores) and missingness (i.e. whether a participant had missing data on any exposure or outcome variables).

**Interaction analyses.** To explore potential interactions between mental health domains and baseline characteristics on weight, we included interaction terms of within-individual and between-individual measurements by, respectively, age, sex, ethnicity, education and baseline BMI in the main models. Interaction terms were entered one at a time. Age and BMI were entered as continuous measures.

To follow up interactions with baseline BMI, we re-fitted the models separately within BMI sub-groups to facilitate interpretation ($<25$kg/m$^2$, 25–29.9kg/m$^2$, $\geq$30kg/m$^2$). The Statistical Analysis Plan stated we would dichotomise into $<25$ and $\geq25$ kg/m$^2$; however on reviewing Fig 1a–1d, it became apparent that the obesity category (30+) is a distinct group and should be examined separately. We did not perform adjustments to control Type I error following recommendations for exploratory observational studies [37].

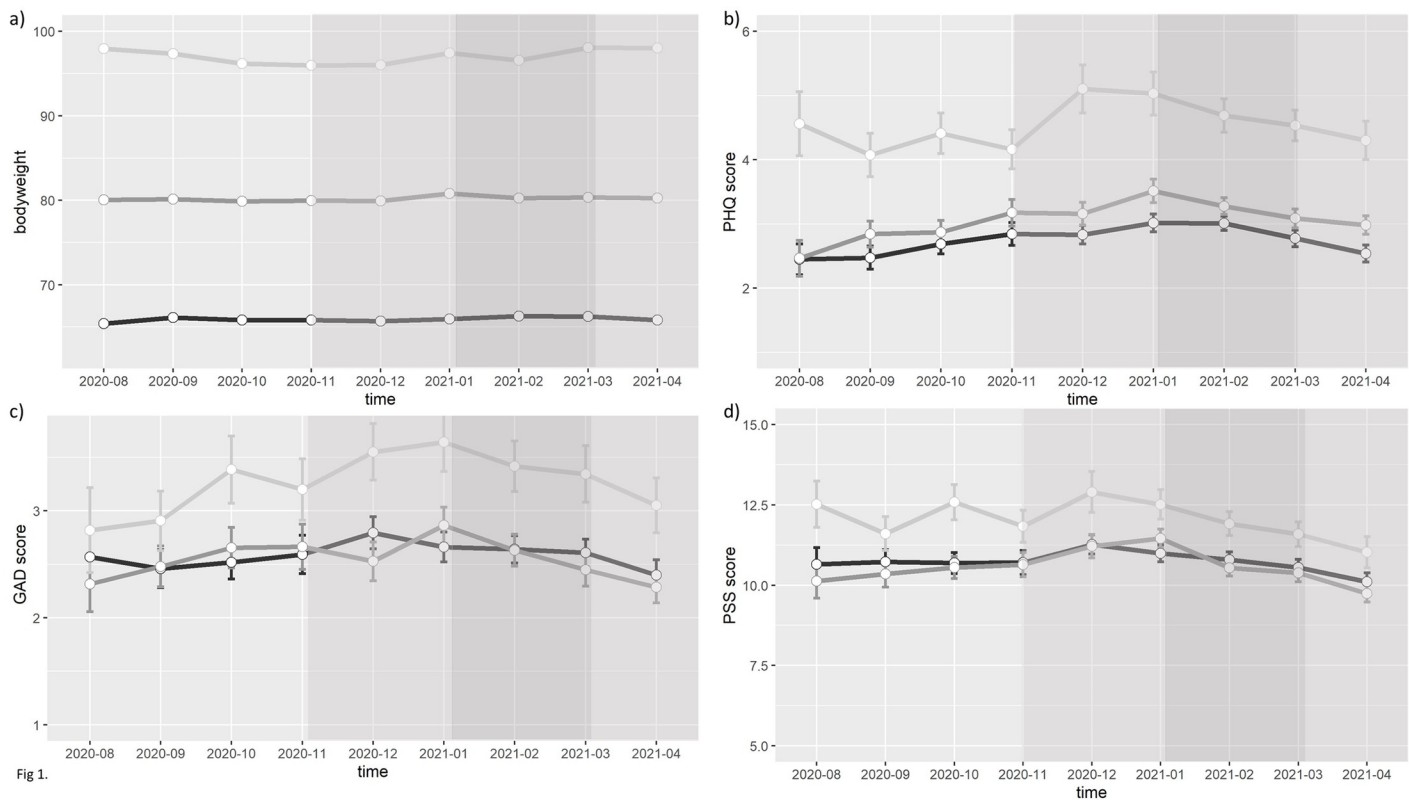

**Fig 1. Body-mass-index (BMI) and mental health scores over the study period, by baseline BMI category.** a) Mean weight, b) depressive symptom score (Patient Health Questionnaire, PHQ-8), c) anxiety symptom score (Generalised Anxiety Disorder, GAD-7), and d) stress score (Perceived Stress Scale, PSS) over the 9-month study period, by baseline BMI category. Error bars are 95% confidence intervals showing within-individual variation. Background shading: Light gray = minimal COVID-19 restrictions, Darker gray = some restrictions, Dark gray = strict restrictions.

## Results

Of the 11,469 Fenland participants that were approached, 4031 participants aged 44–70 years consented to take part in the Fenland COVID-19 study and 2524 of these also consented to participate in the app sub-study. Of these, 2277 participants downloaded the app. We removed weight values that indicated a biologically implausible BMI [26, 27] or biologically implausible weight loss [28]. We excluded participants who reported neither mental health nor weight data (n = 144). Our analysis dataset included 2133 participants. The number of participants who reported mental health and weight measures each month fluctuated between 442 and 1598. On average, participants reported 4.7, 4.8, 4.9 and 5.2 values for stress, depressive symptoms, anxiety symptoms, and weight, respectively, across the study period.

### Descriptive analyses

Those who consented to the app sub-study were, on average, more likely to be younger, male, in the highest socioeconomic status category, in employment, have a university degree, live in an urban area, live with family instead of alone or with friends/partner, and were less likely to live in an area categorised as being deprived than those who did not consent [38]. Baseline characteristics had a similar distribution in those who consented and those included in the analysis (Table 1). PHQ scores above the cut-off indicative of major depression (≥10) were

**Table 1. Sample characteristics of participants who consented to take part in the app sub-study (N = 2524) and who were included in the analyses (N = 2133).** Percentages are calculated using the number of non-missing values as the denominator. PSS = Perceived Stress Score, PHQ = Patient Health Questionnaire, GAD = Generalised Anxiety Disorder questionnaire.

| | N with data | Consented to take part in the app sub-study (N = 2524) | N with data | Included in the analyses (N = 2133) |
|---|---|---|---|---|
| Age in years, Mean (SD) | 2524 | 58.4 (7.0) | 2133 | 58.4 (7.0) |
| Sex: Men, n (%) | 2524 | 1,149 (45.5%) | 2133 | 960 (45.0%) |
| Ethnicity: White, n (%) | 2450 | 2,405 (98.2%) | 2068 | 2031 (98.2%) |
| SES category [a], n (%) | 2443 | | 2062 | |
| Traditional and modern professional and higher managerial | | 1654 (67.7%) | | 1402 (68.0%) |
| Lower managerial and intermediate occupations | | 428 (17.5%) | | 361 (17.5%) |
| Technical/semi-routine and routine occupations | | 361 (14.8%) | | 299 (14.5%) |
| Baseline BMI in kg/m$^2$, Median (IQR) | 2049 | 25.8 (23.2, 29.0) | 1982 | 25.8 (23.3, 29.1) |
| Baseline mental health [b] | | | | |
| Baseline stress (PSS-10), Mean (SD); Median (IQR) | N/A | n/a | 2007 | 11.5 (6.8); 11 (6, 16) |
| Baseline depressive symptoms (PHQ-8), Mean (SD); Median (IQR) | N/A | n/a | 2003 | 3.3 (3.7) 2.0 (0.0, 5.0) |
| Major depression (PHQ $\geq$10), n (%) | N/A | n/a | 2003 | 132 (6.6%) |
| Baseline anxiety symptoms (GAD-7), Mean (SD); Median (IQR) | N/A | n/a | 2015 | 2.8 (3.5); 2.0 (0.0, 4.0) |
| Generalised anxiety disorder (GAD $\geq$10), n (%) | N/A | n/a | 2015 | 94 (4.7%) |

[a] For SES, categories are based on the approach described by Barrett et al. [34].

[b] Based on the first score on mental health questionnaires after entering the study

reported by 132 participants (6.6%), and 94 participants (4.7%) reported scores above the cut-off indicative of generalized anxiety disorder ($\geq$10). The proportion of participants with scores indicating major depression was highest among those with obesity (12.1%, 43/391), followed by those with overweight (5.1%, 37/771), and healthy weight (4.0%, 31/820). For generalised anxiety disorder, the proportions were more similar across BMI groups, with 5.3% (19/391) in those with obesity, 3.6% (26/771) in those with overweight and 4.8% (38/820) in those with healthy weight.

Mean body weight increased slightly in September, January, and March (compared to the preceding month), and these fluctuations were more pronounced in those with higher BMI compared to those with lower BMI (Fig 1a; S1 Table in supplement). Depressive symptoms, anxiety symptoms and perceived stress scores appeared to be generally higher in those with BMI above 30 kg/m$^2$ than in those with BMI below 30 kg/m$^2$ (Fig 1b–1d).

## Missing data

The available sample size differed for each timepoint and each examined association, and varied between 396 and 1050 participants (Table 2). Completion rates were similar for the three mental health questionnaires. Older participants were less likely to have missing data than younger participants (3–5% lower odds of missing data for each year of age). For weight only, women, those with higher BMI, and those with higher baseline mental health scores were more likely to have missing data. Further information on missing data is provided in S2 and S3 Tables in the supplement.

**Table 2. Sample size per month per examined association.**

|  | Aug 2020 | Sep 2020 | Oct 2020 | Nov 2020 | Dec 2020 | Jan 2021 | Feb 2021 | Mar 2021 | Apr 2021 |
|---|---|---|---|---|---|---|---|---|---|
| Weight & stress | 396 | 721 | 824 | 656 | 671 | 915 | 973 | 1049 | 1008 |
| Weight & depressive symptoms | 409 | 743 | 828 | 656 | 696 | 915 | 971 | 1050 | 1006 |
| Weight & anxiety symptoms | 415 | 807 | 900 | 705 | 737 | 928 | 977 | 1049 | 1005 |

## Within-individual variation in mental health and weight

We found no evidence for an association of within-individual variation in either anxiety symptoms or stress with weight at the next assessment timepoint (Table 3). However, there was a positive association between within-individual variation in depressive symptoms and subsequent weight. A within-individual deviation from an individual's median depression score by 1 unit was associated with a weight value that was 0.045kg (0.021 to 0.069kg) higher than the weight that person would have at their median depression level, after adjustment for the covariates. This translates to a within-individual deviation of 5 units from their median depression level (e.g., from a score of 5 = mild symptoms to 10 = moderate symptoms) being associated with a higher weight by approximately 0.23kg (0.105–0.345kg).

To account for any potential violation of the assumption of normality of the residuals, we re-fitted models with the log-transformed outcome and obtained similar results (S4 Table). When examining the reverse direction of association, we found no evidence of an association of within-individual variation in weight with depressive symptoms, anxiety symptoms or stress (S5 Table).

**Table 3. Association of within-individual and between-individual measurements of stress, depression and anxiety with weight over the study period.** All models adjusted for age at baseline, BMI at baseline, sex, education, occupation, restriction level, and seasonality (month). Estimate with confidence interval not including zero highlighted in bold. Between-individual measurements = the median of each person's monthly scores; within-individual measurement = the deviation of each monthly score from that person's median.

|  | Estimated difference in weight (kg) for every unit of the exposure (95% confidence interval) |
|---|---|
| Stress (PSS) |  |
| Between-individual measurement of stress | 0.007 (-0.042 to 0.057) |
| Lagged within-individual measurement of stress | -0.009 (-0.022 to 0.004) |
| Change in raw stress score* | -0.008 (-0.02 to 0.003) |
| Depressive symptoms (PHQ-8) |  |
| Between-individual measurement of depressive symptoms | 0.066 (-0.031 to 0.164) |
| Lagged within-individual measurement of depressive symptoms | **0.045 (0.021 to 0.069)** |
| Change in raw depressive symptom score* | **0.027 (0.006 to 0.047)** |
| Anxiety symptoms (GAD-7) |  |
| Between-individual measurement of anxiety symptoms | 0.017 (-0.090 to 0.123) |
| Lagged within-individual measurement of anxiety symptoms | 0.016 (-0.007 to 0.039) |
| Change in raw anxiety symptom score* | 0.0013 (-0.006 to 0.033) |

* These are the raw summary scores for each scale, i.e., without splitting scores into within-individual and between-individual measurements.

In subgroup analyses, an interaction between within-individual measurement of depressive symptoms and baseline BMI on weight at the next assessment timepoint was observed. In stratified analyses, we found evidence of a dose-dependent moderation effect of baseline BMI on the within-individual association between depressive symptoms and subsequent weight: In those with a BMI of $<25kg/m^2$ at baseline (n = 820), we found no evidence of an association between within-individual variation in depressive symptoms and weight at the next assessment (0.011kg, -0.017 to 0.039kg). However, for those with overweight (BMI 25–29.9kg/m$^2$, n = 771), within-individual variation in depressive symptoms was positively associated with weight, such that a deviation of 1 unit in depressive symptoms from the individual's median score was associated with a higher subsequent weight by 0.052kg (0.010 to 0.094kg). For those with obesity ($>30kg/m^2$, n = 391), a deviation of 1 unit in depressive symptoms from the individual's median score was associated with a higher subsequent weight by 0.071kg (0.013 to 0.129kg). We found no evidence for any other interactions (S6 Table, supplement).

### Between-individual variation in mental health and weight

We found no evidence for an association between either anxiety symptoms, stress or depressive symptoms and weight for between-individual measures (Table 2). Similar results were obtained in models with log-transformed outcomes (S4 Table). We also found no evidence for a reverse direction of association (S5 Table), nor for interactions with any of the baseline variables (S6 Table, supplement).

### Analyses with raw mental health scores

In the models using the raw mental health summary scores (without separation into between- and within-individual measurements), there was a positive association between change in depressive symptoms and change in weight. A 1 unit increase in depressive symptoms between two assessments was associated with a subsequent increase in weight by 0.027kg (0.006 to 0.047) between the next two assessments (Table 3). We found no evidence for a reverse direction of effect (S5 Table), nor for an association between change in stress scores with weight change nor for change in anxiety symptom score with weight change (Table 3).

## Discussion

This was the first longitudinal study to investigate the association between mental health measures (stress, depressive symptoms, anxiety symptoms) and bodyweight with frequent measurements over an extended time period. By measuring mental wellbeing and weight monthly over up to 9 months, we were able to explore how fluctuations in mental wellbeing within individuals (i.e. small changes over time that might not constitute clinical levels of poor mental health) may affect their weight over the next month. We found that within-individual deviations in depressive symptoms from a person's usual depressive symptom level (with the "usual level" defined as the median depressive symptom level) were positively associated with subsequent weight, such that an increase in depressive symptom severity by one unit predicted a weight gain of 0.045kg one month later.

Our findings indicate a dose-dependent moderation effect of baseline BMI: in participants with obesity, the difference in weight associated with a unit difference in depressive symptoms was larger than in those with overweight. In participants with BMI<25kg/m$^2$, we found no evidence for an association. Overall, this suggests that individuals with overweight or obesity are at increased vulnerability to weight gain in response to increases in depressive symptoms. The effect sizes were small, however even small weight changes occurring over short periods of time can lead to larger weight changes in the long-term [39], particularly among those with

overweight and obesity [40]. Apart from BMI, none of the other measured baseline characteristics moderated the relationship between depressive symptoms and subsequent weight, indicating this relationship did not depend on age, sex, education or occupation.

Our findings tentatively suggest that, if weight management interventions for adults with overweight/obesity monitor depressive symptoms over time and intervene when symptoms rise above their usual level, this may help prevent future weight gain. Using mobile phone technology, it is possible to collect data frequently over extended periods of time (as demonstrated in the present study) and use this to provide adaptive, context-specific support [11].

While within-individual deviations in depressive symptoms from each person's usual level predicted subsequent weight, participants' general depressive symptom level compared to others (i.e. between-individual variation) was not associated with weight. Thus, our results suggest that weight management interventions may benefit by considering whether individuals experience changes in their usual depressive symptoms, rather than comparing individuals' levels to others. For example, a particular individual may not score highly on the PHQ-8 compared to others, but this may still indicate a score that is higher-than-usual for that particular individual, and our findings indicate this may predict subsequent weight. However, the present study was an observational, exploratory analysis and findings should be interpreted with caution. The effect estimates for the association of depressive symptoms and weight were small, and we cannot firmly conclude whether there is a direct, causal link. We therefore recommend further investigation prior to implementing changes to weight management interventions.

Previous research suggests that poor mental health is both a cause and consequence of obesity, yet we found evidence that weight did not predict subsequent depressive symptoms. Whilst causality cannot be firmly inferred in this observational study, this indicates the observed relationship between depressive symptoms and weight did not occur due to reverse causality.

Our findings are consistent with the general notion that depressive symptoms are associated with weight. Previous cross-sectional and prospective research has indicated that higher levels of depression are associated with higher weight, BMI and weight-related outcomes such as food intake and reduced physical activity [6, 12–16]. For example, in a longitudinal study, depression predicted higher 7-year increase in BMI (std. $\beta = 0.025$) and waist circumference (std. $\beta = 0.028$) [16]. Our study adds to these findings by exploring repeated short-term measurements rather than extrapolating over longer timeframes, and by elucidating how fluctuations within individuals predict weight one month later. Interestingly, some previous studies have found associations of between-individual variation in depressive symptoms with weight [e.g. 41, 42], whereas we found none. One key difference is that the majority of previous research has involved dichotomising depression (i.e., clinical depression vs. no clinical depression), whereas we used depressive symptoms as a continuous outcome. However, evidence on the association of depression with weight is typically mixed [1] and there may be a multitude of reasons for variations in results. It should be noted that some studies indicate that depression symptoms can be associated with both weight loss and weight gain and increased as well as decreased eating [14, 43, 44], and changes in weight in either direction are symptoms of depression according to the Diagnostic and Statistical Manual of Mental Disorders (DSV-5) [45]. This indicates considerable heterogeneity in this relationship. Prospective repeated-measures studies with separation into between-individual and within-individual effects (such as the present study) can help us to gain a better understanding of such complex relationships. Future studies could consider how different types of depression affect bodyweight [5].

Our study found no evidence of an association between anxiety or stress with subsequent weight nor for an association between weight and subsequent anxiety or stress, neither for the total sample, nor when examining the association by baseline characteristics (age, sex,

education, occupation, BMI). The relationship between negative emotions and eating is complex; negative affect appears to be associated with both increased and decreased food intake [46]. High-intensity or high-arousal emotions such as fear and tension, as opposed to lower-intensity/arousal emotions such as depression, may be more likely to suppress eating [47]. Anxiety and stress may fall into this category. In accordance with this finding, a longitudinal study during the COVID-19 pandemic found that higher anxiety symptoms were associated with lower odds of eating more, and having a stressful life event was associated with eating less [14].

However, it should be noted that, in contrast to our findings, there is also evidence from systematic reviews indicating both stress and anxiety can be associated with higher weight [2–4]. These reviews primarily draw on cross-sectional studies. This may explain why our findings differ.

Additionally, it should be noted that effect sizes of associations between mental health variables and weight tend to be small; an umbrella review and meta-analysis found an effect estimate of Cohen's d = 0.12 (95%CI = 0.09–0.14) [48]. It is therefore possible that our sample size was too small to detect some existing effects, particularly as data completion at the start of the study was initially low (Table 2). The sample size available for the association of weight and stress in Month 1 (n = 396), for example, would have only 39% power to detect an existing effect of d = 0.12. As such we cannot firmly conclude that there is no association between stress and anxiety with weight.

## Strengths and limitations of the present study

This study was strengthened by the inclusion of monthly measures of both the exposures and outcome, allowing a better assessment of temporal associations between mental health and weight than most existing prospective studies which include only annual (or less frequent) assessments. Moreover, the monthly repeated measures allow separation into within-individual and between-individual variation. This allows us to study how changes in individuals' emotional states over time predict weight, and this in turn helps highlight opportunities for early intervention and prevention [11].

It is also worth noting that many studies categorise variables of affect and weight based on clinical cut-offs, whereas our study treated these variables as continuous. Treating mental health as a symptom continuum (rather than dichotomising into those with/without a diagnosis) appreciates that individuals can experience one or more symptoms of mental illness without meeting diagnostic criteria, and this can have a meaningful impact on health and wellbeing. As such, our study gives an indication of how small variations in depressive symptoms, anxiety symptoms and stress relate to small changes in weight. This means our findings are relevant to the general population rather than pertaining only to those with clinical levels of poor mental health. Studying mental health in this way can provide more detailed insights for public health practice and policy.

Our approach of using each participant's deviation from their median mental health score to represent within-individual variation will work less well in participants with fewer mental health scores and in people with large variations in mental health across the study period, since the median may then be less representative of the individual's general mental health level (or perhaps some participants do not have a "general mental health level"). For this reason, we repeated the analysis using the change in raw mental health summary scores between assessment timepoints. The analysis confirmed our findings, indicating that changes in stress and anxiety symptoms were not associated with subsequent weight change, but changes in

depressive symptom level were predictive of weight changes (and no evidence was found for a reverse direction of effect).

Data in the present study are derived from a regional cohort of adults aged 44–70 years when recruited into the COVID-19 study and, although they are broadly representative of Cambridgeshire, our results may not be generalisable to other populations with different socio-demographic characteristics. Additionally, higher age was associated with 3–5% lower odds of having missing data on exposures and the outcome. Our findings may, therefore, be more applicable to older participants. For weight, men, those with lower BMI, and those with better mental health were less likely to have missing data and were likely to provide more measurements. Our findings should, therefore, be interpreted with caution. Future studies should carefully consider how to increase reporting of weight among women, those with higher BMI, and those with poorer mental health outcomes.

This study involved conducting several inferential statistical tests. As such, there is a risk of finding p-values below 0.05 simply due to the number of tests performed. Adjustments to control for Type I error such as the Bonferroni adjustment bear the risk of inflating Type II error. Bender & Lange [37] recommend that application of multiplicity adjustments should be reserved for confirmatory trials, rather than exploratory analyses of observational data. We therefore did not perform any adjustments. Or results should be interpreted with caution. Although we cannot infer a causal relationship between depressive symptoms and weight, the temporal association and lack of evidence for a reverse direction of association (i.e., depressive symptoms predicted subsequent weight, but weight did not predict depressive symptoms) lends some credence to the idea that within-individual changes in depression symptoms influence weight. However, it should be noted that, when examining an association between an exposure variable and an outcome, the regression coefficient will be unbiased (but have wider confidence intervals) if there is random measurement error in the outcome. However, random measurement error in the exposure will bias the regression coefficient towards zero [49]. Therefore, if weight had high random measurement error, this may have biased a true effect of weight on depressive symptoms towards the null. We used self-reported weight since this study was conducted during the COVID-19 pandemic with social distancing measures in place. Self-reported weight could also involve systematic measurement error (e.g., people with high/low depressive symptom levels may systematically over- or under-report their weight). This could create a spurious association between depressive symptoms and weight. Nevertheless, findings from other studies have shown that self-reported weight is valid and reproducible, with a correlation of 0.9 with measured weight [50].

It is also possible that a third, unrelated variable influences both depressive symptoms and weight. We controlled for several potential confounders (including sociodemographic variables, baseline BMI, seasonality, and COVID-19 restrictions), and the within-individual analysis further addresses potential time-invariant confounding. Physical activity could also be an important related variable, particularly in light of COVID-19 restrictions. In an observational study during the COVID-19 pandemic, higher BMI was associated with lower reported levels of physical activity [51], and physical activity is linked with depressive symptoms [52]. Thus, one alternative explanation of our findings is that changes in physical activity caused both changes in depressive symptoms and weight.

## Conclusions

Findings from this prospective, longitudinal study during the COVID-19 pandemic indicate that, when an individual's depressive symptom level rises compared to their usual level, this predicts weight gain within the next month. Those who are already at risk for poor health

outcomes due to their increased BMI are most likely to experience weight gain, potentially leading to further health deteriorations. This suggests that monitoring and addressing depressive symptoms in individuals with overweight or obesity may aid weight gain prevention. While clinical depression should be treated regardless of whether this is associated with weight gain, the present study examines small changes in depressive symptoms, recognizing that individuals may experience meaningful fluctuations in mental wellbeing that do not indicate a clinical diagnosis. Our findings suggest weight management interventions may benefit by monitoring for small fluctuations in depressive symptoms and providing additional emotional support when needed. Thus, our findings support a more holistic approach to weight management, taking changes in individuals' mental wellbeing over time into account.

## Supporting information

**S1 Table. Mean weight and mean mental health scores per month over the study period (August 2020 –April 2021).** PSS = Perceived Stress Score, PHQ = Patient Health Questionnaire, GAD = Generalised Anxiety Disorder questionnaire.
(DOCX)

**S2 Table. Outcome completion rates.** Number and percentage (out of those who downloaded and onboarded to the app) of participants providing data per month per variable (stress, depression, anxiety, weight). Note: Participants were only asked to begin completing mental health questionnaires ~30 days after onboarding; therefore completion rates in August are low compared to subsequent months.
(DOCX)

**S3 Table. Association between baseline characteristics and missingness (i.e., whether the participant had at least one missing data point for the respective outcome at any time-point) using logistic regression.** Estimates of odds ratios with confidence intervals not including 1 are marked in bold. PSS = Perceived Stress Score, PHQ = Patient Health Questionnaire, GAD = Generalised Anxiety Disorder questionnaire.
(DOCX)

**S4 Table. Regression models for log-transformed outcomes.** Association between, respectively, stress, depressive symptoms and anxiety symptom scores and log-transformed body-weight over the study period. All models adjusted for age at baseline, BMI at baseline, sex, education, occupation, restriction level (alternative categorisation), and seasonality (month). Note regression coefficients cannot be interpreted in terms of difference in bodyweight because the outcome is log-transformed. Estimates with confidence intervals not including zero are marked in bold. PSS = Perceived Stress Score, PHQ = Patient Health Questionnaire, GAD = Generalised Anxiety Disorder questionnaire.
(DOCX)

**S5 Table. Regression models for reverse direction of association.** Association of within-individual and between-individual measurements of weight with stress, depressive symptoms and anxiety symptoms over the study period. All models adjusted for age at baseline, BMI at baseline, sex, education, occupation, restriction level, and seasonality (month). Estimate with confidence interval not including zero highlighted in bold. Between-individual measurements = the median of each person's monthly scores; within-individual measurement = the deviation of each monthly score from that person's median.
PSS = Perceived Stress Score, PHQ = Patient Health Questionnaire, GAD = Generalised

Anxiety Disorder questionnaire.
(DOCX)

**S6 Table. Sub-group analyses.** Relationship between mental health variables and bodyweight by sociodemographic variables and baseline BMI. In addition to the interaction terms, the model includes the following fixed effects: the main effects of the interaction variables, time, season (month as dummy), restriction level, age, sex, baseline BMI. Estimates with confidence intervals not including zero are marked in bold. PSS = Perceived Stress Score, PHQ = Patient Health Questionnaire, GAD = Generalised Anxiety Disorder questionnaire.
(DOCX)

**S1 Checklist. STROBE statement—Checklist of items that should be included in reports of *cohort studies*.**
(DOC)

## Acknowledgments

We are grateful to the Fenland Study participants for their willingness and time to take part. We thank all members of the following teams responsible for practical aspects of the study; Study Coordination, Field Epidemiology, Anthropometry Team, Physical Activity Technical Team, IT and Data Management.. KR and NJW designed the study. GMK contributed to study design. JM, KR, ALA and SJS conceptualised and wrote the analysis plan. EDLR contributed to data management including processing and analysis. JM analysed and interpreted the data and wrote the first draft of the manuscript. AD supported the analysis and the creation of data visualisations. All authors provided critical review for intellectual content before submission and approved the final manuscript. The views expressed are those of the author(s) and not necessarily those of the NHS, the NIHR or the Department of Health and Social Care.

## Author Contributions

**Conceptualization:** Julia Mueller, Amy L. Ahern, Stephen J. Sharp, Arabella Zuckerman, Benjamin I. Perry, Golam M. Khandaker, Kirsten L. Rennie.

**Data curation:** Julia Mueller, Emanuella De Lucia Rolfe.

**Formal analysis:** Julia Mueller.

**Funding acquisition:** Nick J. Wareham.

**Investigation:** Kirsten L. Rennie.

**Methodology:** Julia Mueller, Amy L. Ahern, Stephen J. Sharp, Benjamin I. Perry, Golam M. Khandaker, Emanuella De Lucia Rolfe, Kirsten L. Rennie.

**Supervision:** Amy L. Ahern, Rebecca A. Jones, Stephen J. Sharp, Benjamin I. Perry, Nick J. Wareham, Kirsten L. Rennie.

**Visualization:** Alan Davies.

**Writing – original draft:** Julia Mueller.

**Writing – review & editing:** Julia Mueller, Amy L. Ahern, Rebecca A. Jones, Stephen J. Sharp, Alan Davies, Arabella Zuckerman, Benjamin I. Perry, Golam M. Khandaker, Emanuella De Lucia Rolfe, Nick J. Wareham, Kirsten L. Rennie.

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
