## [Decision Letter · Decision Letter 0]

11 May 2023

PONE-D-23-00921

The relationship of within-individual and between-individual variation in mental health with bodyweight: A longitudinal study

PLOS ONE

Dear Dr. Mueller,

Thank you for submitting your manuscript to PLOS ONE. After careful consideration, we feel that it has merit but does not fully meet PLOS ONE’s publication criteria as it currently stands. Therefore, we invite you to submit a revised version of the manuscript that addresses the points raised during the review process.

In particular, please pay attention to Reviewer 1's feedback on the use of median scores.

We look forward to receiving your revised manuscript.

Kind regards,

Abigail Emma Russell

Academic Editor

PLOS ONE

Journal Requirements:

"We are grateful to the Fenland Study participants for their willingness and time to take part. We thank all members of the following teams responsible for practical aspects of the study; Study Coordination, Field Epidemiology, Anthropometry Team, Physical Activity Technical Team, IT and Data Management. Measurement of anthropometry was supported by the NIHR Cambridge Biomedical Research Centre (IS-BRC-1215-20014). "

"The study was supported by the Medical Research Council (grant MC_UU_00006/1). The funder had no role in study design, data collection and analysis, decision to publish, or preparation of the manuscript."

"JM is a trustee for the Association for the Study of Obesity (unpaid role). GMK receives royalties from Cambridge University Press (Editor of Textbook of Immunopsychiatry). ALA is the chief investigator on two publicly funded (MRC, NIHR) trials where the intervention is provided by WW (formerly Weight Watchers) at no cost outside the submitted work. ALA is on the Scientific Advisory Board for WW (payment to institution). The views expressed are those of the author(s) and not necessarily those of the NIHR, MRC or other funders. Huma Therapeutics Ltd. developed, validated and tested the app at no charge."

Reviewers' comments:

Reviewer's Responses to Questions

**Comments to the Author**

1. Is the manuscript technically sound, and do the data support the conclusions?

Reviewer #1: Yes

Reviewer #2: Yes

Reviewer #3: Yes

2. Has the statistical analysis been performed appropriately and rigorously? 

Reviewer #1: Yes

Reviewer #2: I Don't Know

Reviewer #3: Yes

3. Have the authors made all data underlying the findings in their manuscript fully available?

Reviewer #1: No

Reviewer #2: No

Reviewer #3: Yes

4. Is the manuscript presented in an intelligible fashion and written in standard English?

Reviewer #1: Yes

Reviewer #2: Yes

Reviewer #3: Yes

5. Review Comments to the Author

Reviewer #1: This is an interesting study that uses a novel approach to explore how changes in mental health relate to changes in body weight. I think the work will make a useful addition to the literature. Below are some comments I believe need to be addressed before publication:

Abstract – the Conclusions section for me is too far removed from the study. This is a small preliminary observational study with limitations, so rather than attempting to conclude about weight management interventions I think it is more appropriate to make conclusions regarding what the study actually investigated.

The introduction is well written, the methods are reported well.

Analysis approach. I was not overly convinced about using median mental health score as a relative value (e.g., how much does depression for a month differ from that person’s median monthly depression score), as opposed to just using change in mental health (e.g. extent to which depression changes from one month to the next). With the median value it suffers from the median being drawn from a person’s mental health across the entire study. As an example, using hypothetical data I show below of 5 months of depression data, this person’s depression scores go down by 2 points across the first two months, whereas a median relative change would compute the change from month 1 to month 2 as 50% smaller. Likewise from month 4 to 5 the person has a sharp increase in depression, but a median relative change (difference of 4) characterises this as being 20% smaller than an absolute change (5).

Month 1 = 3, month 2 = 1, month 3 = 2, month 4= 1, month 5 = 6

Median = 2

I also wonder how things play out using a median approach when a person has very chaotic monthly mental health variation. A median assumes that everyone has a ‘baseline’ value, but maybe over such a relatively short study period this isn’t so much the case. The other question is can you report on what the number of observations median values are typically derived from? i.e. is it the case that some pps only have a few data points across waves, so their median may not be at all representative of their true ‘baseline’ depression in normal life.

Theoretically the median approach also doesn’t seem to fit what one would expect. Presumably it’s decrease or increase in recent mood that impacts on my motivation and behaviour, as opposed to me monitoring what my current mood is relative to my median? And finally, the median feels somewhat artificial because we are making inferences at times when a median doesn’t yet exist (e.g., to what extent does depression in month 1 differ from the median).

I’m interested to hear what the authors have to say on all of the above and I’d also want to see what results look like when mental health variables / body weight are treated in a more traditional cross-lagged approach (i.e. looking at how change from one wave to the next predicts outcomes), as this would be theoretically informative. Maybe it produces the exact same results, but either way it’d be informative.

Sample size reporting. Although missing data is mentioned in the supps, I think the manuscript would benefit from providing a summary of this in the text and also make it clear what the ramifications of this are in terms of the analytic sample size, i.e. on a wave by wave basis provide some information about how many pps had all the available data, or whatever. At present its difficult to workout how many pps had monthly change data for analyses at each wave or overall how many months of change pps contributed to on average etc.

Analytic sample size justification. The reason the above seems relevant and needed is because there is no justification for the analytic sample size as being sufficiently large enough for the purposes of the study and analysis approach (i.e. statistical power for models used and number of pps in these models). Some attempt to examine how well powered the analyses are is needed. Dependent on what this looks like, one thing to consider is that the psychological correlates of body weight tend to be statistically small - https://pubmed.ncbi.nlm.nih.gov/33086131/ so it may be the case that a limitation of the present analyses is sample size.

Missing data. Can the authors explain how missing data is treated in the analyses.

Multiple comparisons. There are a lot of analyses. The p value appears to be set a 0.05. there is therefore a high risk of false positives (i.e. maybe the effect observed is one of these). If the authors have adjusted for this already then they could make this clearer. If they haven’t, then some justification as to why not would be needed and heavy caveating of results, the study being ‘exploratory’ and so forth.

The authors measure potential 3rd variables. One that seems highly plausible is exercise. In work we did during COVID - https://pubmed.ncbi.nlm.nih.gov/33038479/ we found that people of heavier weight were more likely to report exercising less. We know exercise and depression are linked. Therefore, one potential alternative explanation of the present findings would be that the reason depression and weight covary is not because depression is predicting/causing weight gain but rather a decrease in exercise is causing both.

Page 21. Measurement error biasing a true effect to the null. I don’t fully follow this – isn’t is feasible that it can bias the other wat – i.e. the more depressed I am the more likely I am to report my weight in a way that is negative (heavier) and vice versa – the happier I am the more likely I am to report with rose tinted glasses….

Reviewer #2: Dear authors,

This was an interesting paper looking at not only the relationship of mental health and weight across people, but also at mental health as predictor for weight within people. The only significant relationship found was depression variance as predictor for weight, and the moderating effect of BMI on this relationship.

I think, this study adds to the current literature, but would recommend minor changes:

Abstract:

Results: Some of the phrasing is harder to follow without the information in the main text and therefore makes it hard to understand the abstract in itself, e.g., what do you mean by ‘within-individual variation’ and ‘dose-dependent’ (I wouldn’t call BMI categories ‘doses’). While it is accurate, for the abstract it might be helpful to describe it in easier terms which are more commonly used.

CI intervals are sometimes written in square brackets and sometimes not. Should be uniform.

You mention the reverse direction of association in the results part. It was not mentioned before that you are interested in it and therefore, comes a bit surprisingly.

Background:

l. 92: This sentence could be clearer.

l. 94-100: You mention that COVID gives a unique opportunity to assess the changes, as these variations in mental health are stronger than usual. But could not COVID influence weight via other pathways, e.g., increased sedentary behaviour? You took restrictions into account, but the effect were sometimes extending these restriction periods and would therefore not be covered by the covariate of restrictions.

l. 107-109 (also related to l.116): How did you define the ‘usual levels’ of mental health? Is that the baseline or mean? Is this difference between their measured level to the usual level the ‘within-individual variation’ you mention in aim i)?

l.122-123: By ‘these associations’, do you mean all three aims you presented before or just specific ones?

Is this study/these analyses exploratory or confirmatory? If they were confirmatory, you should add hypotheses. If they were exploratory, it would be good to mention this briefly.

Methods:

As you want to use depressive symptoms as predictive marker for weight gain, it might be useful to look at the relationship between changes in depressive symptoms and weight change.

Why did you choose a 1-month gap between mental health and weight measure?

As you use baseline BMI and age from the main Fenland study as covariates, it would be of interest when the main Fenland study was conducted or how much time was between the main Fenland study and the COVID app sub-study. This helps to get a feeling how BMI could have changed.

l.198-199: In a later table you mention that the occupation levels were taken from literature. It might be useful to include this information also in the confounder section.

l. 228-230: You label the within-individual measures of mental health ‘lagged’, but from the description it is the measure before all the other measures, not after. ‘Lagged’ has more the meaning of ‘later’, so I find it confusing. Please also clarify if all variables (such as timepoint, between-individual measurements, season and weight) are from one timepoint and only lagged within-individual measurements from the timepoint before.

l.240-241: What would be the sub-groups?

l.239: Clarify which analysis you used. Was another random intercepts regression model?

l.245-246: Could be clearer.

You sometimes mention a statistical analysis plan. Did you pre-register the plan somewhere? Then please add a link to the publication.

Results:

While some information is interesting, it might not be needed in the main paper but could be added to the supplemental data, such as the descriptives of people who consented in taking part in the app-study and missing data.

l.252-255: It could be clearer that the exclusion of people was from the 2524 who consented to take part, not the 2277 who downloaded the app.

Variables in tables should all be uniformly written, e.g., in S1 variables are written in all capital except anxiety. Overall, style, font size, and font should be consistent across all tables and figures. Are page numbers needed in supplementals?

The figures could be more ‘professional’, e.g., figure 1 the label for the legend is ‘bmi_cat’ which could be nicer with ‘BMI cat’ or even ‘BMI categories’ if space permits. Even abbreviation is used, it should be explained. For figure 2, the labels ‘within-ind. Variation…’ should look the same as the ‘weight’ label.

Figure 2: What does the black line with triangles show? The distribution of the weight on the axis could be more equal. I am aware that people are in different BMI categories and therefore the weight is different, but to be able to compare changes, the distance might be good to keep equal, e.g., 0.5kg

l.290: The supplemental data here are S5 and 6, but afterwards you have the data S1-4. It might be more logical to have the supplemental data in order of appearance in the text.

S6: The label at the separate document and the pdf manuscript are not the same. Usually, you explain that values in bold are CI who do not include 0; this is missing here.

l 291-292: You explain how many of the 2277 who downloaded the app did not provide data. Would you not exclude these participants as you would need mental health data for your analyses? Then the information would be more appropriate in the participants section and the final number of participant data would be reduced. If you still included it as participants gave weight and other data, it might be more useful to know how many of the 2133 participants which you included in your analyses did provide no mental health data at all.

l. 301: ‘weight value of 0.045kg’, ‘of’ missing

Discussion:

l. 389-391: You say here that your research is in line with previous findings from cross-sectional studies that depression is related to weight. However, when you analyse the association of between-individual depression and weight, it is almost like a cross-sectional study. But you didn’t find a relationship in the between-individual differences and weight. Therefore, you do not support the results of previous cross-sectional studies. Could you explain why you did not find a relationship. Also in context of finding an association of within-individual variation in depression.

l. 420-431: You explain in this paragraph that previous literature showed that stress and anxiety were related to higher weight. As you already tried to explain your findings in the previous section (which are in contrast with the literature highlighted in this paragraph), it is not clear how the results of this paragraph are related to your study. You mention that longitudinal studies such as yours are missing, but I miss the link to the previous section and your results. Why do you think more longitudinal studies are needed when your findings contradict previous studies?

Reviewer #3: This is an excellent study that is extremely welcome within the literature – it reflects a valuable longitudinal dataset which is a rarity. The question asked is extremely topical. It is a thorough manuscript that is written with clarity. I thought the way that unit change in depression impacted weight was really well explained. I just have four points that the authors might consider.

1. The quality of the figures is noticeably quite low (or at least how they have come out in the pdf I have downloaded) – I would highly recommend finding a way to improve the quality.

2. It is good to see that the small effect sizes are reflected on in the discussion section and as I mentioned before the way relationship between the predictor and the outcome is well explained. It would be helpful to have an understanding of the variance of the dependent variables explained by the models (R-squared value? – apologies if it is in there and I missed it).

3. Did you look at the extent to which across time, rise in depression score led to missing values the following month?

4. Finally, was this study pre-registered in line with open research practice? This would be an ideal study to pre-register in order to increase rigour as this kind of dataset could be vulnerable to p-hacking…

6. PLOS authors have the option to publish the peer review history of their article (what does this mean?). If published, this will include your full peer review and any attached files.

Reviewer #1: **Yes: **eric robinson

Reviewer #2: No

Reviewer #3: **Yes: **Laura Wilkinson

---

## [Author Response · Author response to Decision Letter 0]

3 Aug 2023

Please note, all pages and line numbers refer to the document with tracked changes. 

Reviewer #1

This is an interesting study that uses a novel approach to explore how changes in mental health relate to changes in body weight. I think the work will make a useful addition to the literature. Below are some comments I believe need to be addressed before publication:

Thank you very much for taking the time to review our manuscript and for the constructive, insightful comments. We really appreciate it. 

Comment 1

Abstract – the Conclusions section for me is too far removed from the study. This is a small preliminary observational study with limitations, so rather than attempting to conclude about weight management interventions I think it is more appropriate to make conclusions regarding what the study actually investigated.

Response 1

We have edited the conclusions section in the abstract. It now reads:

“In this exploratory study, individuals with overweight or obesity were more vulnerable to weight gain following higher-than-usual (for that individual) depressive symptoms than individuals with a BMI<25kg/m2.” (p. 2, lines 54-56)

Comment 2

The introduction is well written, the methods are reported well.

Analysis approach. I was not overly convinced about using median mental health score as a relative value (e.g., how much does depression for a month differ from that person’s median monthly depression score), as opposed to just using change in mental health (e.g. extent to which depression changes from one month to the next). With the median value it suffers from the median being drawn from a person’s mental health across the entire study. As an example, using hypothetical data I show below of 5 months of depression data, this person’s depression scores go down by 2 points across the first two months, whereas a median relative change would compute the change from month 1 to month 2 as 50% smaller. Likewise from month 4 to 5 the person has a sharp increase in depression, but a median relative change (difference of 4) characterises this as being 20% smaller than an absolute change (5).

Month 1 = 3, month 2 = 1, month 3 = 2, month 4= 1, month 5 = 6

Median = 2

I also wonder how things play out using a median approach when a person has very chaotic monthly mental health variation. A median assumes that everyone has a ‘baseline’ value, but maybe over such a relatively short study period this isn’t so much the case. The other question is can you report on what the number of observations median values are typically derived from? i.e. is it the case that some pps only have a few data points across waves, so their median may not be at all representative of their true ‘baseline’ depression in normal life.

Theoretically the median approach also doesn’t seem to fit what one would expect. Presumably it’s decrease or increase in recent mood that impacts on my motivation and behaviour, as opposed to me monitoring what my current mood is relative to my median? And finally, the median feels somewhat artificial because we are making inferences at times when a median doesn’t yet exist (e.g., to what extent does depression in month 1 differ from the median).

I’m interested to hear what the authors have to say on all of the above and I’d also want to see what results look like when mental health variables / body weight are treated in a more traditional cross-lagged approach (i.e. looking at how change from one wave to the next predicts outcomes), as this would be theoretically informative. Maybe it produces the exact same results, but either way it’d be informative.

Response 2

Thank you for this insightful comment. We found it really helpful in thinking in more detail about our analyses and our findings and what they might mean. 

We appreciate that, in cases where participants have only a small number of (or very ‘chaotic’) observations, the median approach may work less well. We therefore fully agree that it would be helpful and important to also provide the outputs of the more traditional models, using change in the raw values for depressive symptoms from one timepoint to the next. However, we also still feel that there is value in using the median approach and we have therefore opted to retain these findings (but supplement them with the “traditional” models to help the reader interpret the findings). 

As we explain in the background section, we were particularly interested in separating within-individual fluctuation (how does the individual’s current level compare to their usual level) and between-individual fluctuation (how high is a person’s general depressive symptom level compared to the general level of others). We were interested in this because “Associations at the between-individual and the within-individual level can differ in both size and direction”. 

In the “traditional” model using change in the raw depressive symptom scores, the effect estimate for the association between depressive symptoms and weight includes both within-individual and between-individual variation and it is not possible to separate these out. We therefore decided to use the two different measures of depressive symptoms (their median value = their “trait” depression value, and the deviation from the median at each timepoint = their “state” depression value) for each participant to give separate effect estimates. While we recognise that this method may, in cases (e.g., participants with few observations), be imperfect, we nevertheless argue that this is a useful and novel insight into the relationship between depressive symptoms and weight. 

To illustrate this using the example provided by the reviewer: While this participant shows a marked increase in depressive symptoms from month 4 to month 5 (from 1 to 6), we would argue that it is also useful to look at the difference to the median. The median of 2 reflects that this participant’s depressive symptom level is not always as low as 1; it does fluctuate and at other timepoints the level is at 2 or 3. Hence, while there is definitely an increase in depressive symptoms from month 4 to 5, it is not as stark for that participant as might be assumed when looking only at the values at month 4 and 5, without taking the other values into account. Taking the median into account gives us a better indication of how ‘unusual’ this fluctuation is, considering all the measured values. Thus, the difference to the median gives a better indication of how the value at month 5 differs from the participant’s overall level than the absolute comparison from month 4 to month 5. 

This example shows that we can gain different (but equally useful) insights into the participant’s changes in depressive symptoms using the two different approaches. 

However we do recognise that it would be helpful to a) supplement our analyses with the “traditional” models and b) explain the limitations in the discussion section better so that the reader can interpret the findings and c) be very clear at all times exactly what we are reporting on to allow the reader to interpret the findings.

In light of this, we have made the following changes to the manuscript: 

a) Supplement the analyses

We re-fit our models using change in the raw mental health scores from one timepoint to the next and found a positive association between change in depressive symptoms and change in weight. We have added the following to the methods section:

“Additionally, to provide further context for the interpretation of the findings, we also fitted random intercept regression models for the association between change in mental health scores and change in weight (and the reverse), without splitting up the exposure variables into “within-individual” and “between-individual” variation. In these models, we regressed change in mental health scores from one timepoint to the next onto change in weight between the next timepoints (e.g., change in mental health score from month 1 to month 2 onto change in weight from month 2 to month 3).” (p. 12, lines 256-262)

Also, in response to the reviewer’s request for us to report on the number of observations median values are typically derived from, we have added the following to the results:

“On average, participants reported 4.7, 4.8, 4.9 and 5.2 values for stress, depressive symptoms, anxiety symptoms, and weight, respectively, across the study period.” (p. 14, lines 288-289)

We have added the following to the Results:

“In the models using the raw mental health summary scores (without separation into between- and within-individual measurements), there was a positive association between change in depressive symptoms and change in weight. A 1 unit increase in depressive symptoms between two assessments was associated with a subsequent increase in weight by 0.027kg (0.006 to 0.047) between the next two assessments (Table 3). We found no evidence for a reverse direction of effect (Table S5), nor for an association between change in stress scores with weight change nor for change in anxiety symptom score with weight change (Table 3).” (p. 19, lines 391-397)

b) Explain the limitations

In the discussion, we have added:

“Our approach of using each participant’s deviation from their median mental health score to represent within-individual variation will work less well in participants with fewer mental health scores and in people with large variations in mental health across the study period, since the median may then be less representative of the individual’s general mental health level (or perhaps some participants do not have a “general mental health level”). For this reason, we repeated the analysis using the change in raw mental health summary scores between assessment timepoints. The analysis confirmed our findings, indicating that changes in stress and anxiety symptoms were not associated with subsequent weight change, but changes in depressive symptom level were predictive of weight changes (and no evidence was found for a reverse direction of effect).” (p. 24-25, lines 525-534) 

c) Be clear exactly what we are reporting on

We have made edits in the discussion to be clearer on what we are reporting: 

“We found that within-individual deviations in depressive symptoms from a person’s usual depressive symptom level (with the “usual level” defined as the median depressive symptom level) were positively associated with subsequent weight” (p. 19-20, lines 404-407)

Comment 3

Sample size reporting. Although missing data is mentioned in the supps, I think the manuscript would benefit from providing a summary of this in the text and also make it clear what the ramifications of this are in terms of the analytic sample size, i.e. on a wave by wave basis provide some information about how many pps had all the available data, or whatever. At present its difficult to workout how many pps had monthly change data for analyses at each wave or overall how many months of change pps contributed to on average etc.

Response 3

A summary of information on missing data is provided on p. 16, lines 324 – 333. Additionally, to provide further information on the analytic sample size, we have now added Table 2 to the manuscript (p. 16), which details how many participants provided data for each outcome at each timepoint.

“The available sample size differed for each timepoint and each examined association, and varied between 396 and 1050 participants (Table 2). Completion rates were similar for the three mental health questionnaires. Older participants were less likely to have missing data than younger participants (3-5% lower odds of missing data for each year of age). For weight only, women, those with higher BMI, and those with higher baseline mental health scores were more likely to have missing data. Further information on missing data is provided in Tables S42-S35 in the supplement.”

Comment 4

Analytic sample size justification. The reason the above seems relevant and needed is because there is no justification for the analytic sample size as being sufficiently large enough for the purposes of the study and analysis approach (i.e. statistical power for models used and number of pps in these models). Some attempt to examine how well powered the analyses are is needed. Dependent on what this looks like, one thing to consider is that the psychological correlates of body weight tend to be statistically small - https://pubmed.ncbi.nlm.nih.gov/33086131/ so it may be the case that a limitation of the present analyses is sample size.

Response 4

Thank you for signposting this very interesting and relevant umbrella review. We have added the following to our discussion:

“Additionally, it should be noted that effect sizes of associations between mental health variables and weight tend to be small; an umbrella review and meta-analysis found an effect estimate of Cohen’s d = 0.12 (95%CI=0.09-0.14) (45). It is therefore possible that our sample size was too small to detect some existing effects, particularly as data completion at the start of the study was initially low (Table 2). The sample size available for the association of weight and stress in Month 1 (n=396), for example, would have only 39% power to detect an existing effect of d=0.12. As such we cannot firmly conclude that there is no association between stress and anxiety with weight.” (p. 23-24, lines 499-506)

Comment 5

Missing data. Can the authors explain how missing data is treated in the analyses.

Response 5

We have added the following:

“Random intercept models use all available data and assume that missing values are missing at random (MAR).” (p. 12, lines 263-264)

Comment 6

Multiple comparisons. There are a lot of analyses. The p value appears to be set a 0.05. there is therefore a high risk of false positives (i.e. maybe the effect observed is one of these). If the authors have adjusted for this already then they could make this clearer. If they haven’t, then some justification as to why not would be needed and heavy caveating of results, the study being ‘exploratory’ and so forth.

Response 6

Thank you; this is an important point. We have added a section to the analysis section and to the discussion to clarify that we did not undertake adjustments to control for Type 1 error and why:

Statistical Analyses section:

“We did not perform adjustments to control Type I error following recommendations for exploratory observational studies [38].” (p. 13, lines 277-278)

Discussion:

“This study involved conducting several inferential statistical tests. As such, there is a risk of finding p-values below 0.05 simply due to the number of tests performed. Adjustments to control for Type I error such as the Bonferroni adjustment bear the risk of inflating Type II error. Bender & Lange [38] recommend that application of multiplicity adjustments should be reserved for confirmatory trials, rather than exploratory analyses of observational data. We therefore did not perform any adjustments. Our results should be interpreted with caution.” (p. 25, lines 545-550)

We have also edited our discussion (and abstract) to make it clearer to the reader that this study constitutes an exploratory analysis that should be interpreted with caution, and that, while we outline some potential implications, these warrant further investigation prior to implementation in practice:

“Our findings tentatively suggest that, if weight management interventions for adults with overweight/obesity monitor depressive symptoms over time and intervene when symptoms rise above their usual level, this may help prevent future weight gain.” (p. 20, line 420)

“However, the present study was an observational, exploratory analysis and findings should be interpreted with caution. The effect estimates for the association of depressive symptoms and weight were small, and we cannot firmly conclude whether there is a direct, causal link. We therefore recommend further investigation prior to implementing changes to weight management interventions.” (p. 21, lines 434-438)

“Thus, our results suggest that…” (instead of: “Thus, our results highlight that…”) (p. 20, line 429)

Abstract: “In this exploratory study, individuals with overweight or obesity were more vulnerable to weight gain…” (p. 2, line 54)

Comment 7

The authors measure potential 3rd variables. One that seems highly plausible is exercise. In work we did during COVID - https://pubmed.ncbi.nlm.nih.gov/33038479/ we found that people of heavier weight were more likely to report exercising less. We know exercise and depression are linked. Therefore, one potential alternative explanation of the present findings would be that the reason depression and weight covary is not because depression is predicting/causing weight gain but rather a decrease in exercise is causing both.

Response 7

Thank you, this is a valid point that warrants highlighting in the manuscript. We have added the following to the section where we discuss that our findings could be explained by a third variable:

“It is also possible that a third, unrelated variable influences both depressive symptoms and weight. We controlled for several potential confounders (including sociodemographic variables, baseline BMI, seasonality, and COVID-19 restrictions), and the within-individual analysis further addresses potential time-invariant confounding. Physical activity could also be an important related variable, particularly in light of COVID-19 restrictions. In an observational study during the COVID-19 pandemic, higher BMI was associated with lower reported levels of physical activity (52), and physical activity is linked with depressive symptoms (53). Thus, one alternative explanation of our findings is that changes in physical activity caused both changes in depressive symptoms and weight.” (p. 26, lines 568-576)

Comment 8

Page 21. Measurement error biasing a true effect to the null. I don’t fully follow this – isn’t is feasible that it can bias the other wat – i.e. the more depressed I am the more likely I am to report my weight in a way that is negative (heavier) and vice versa – the happier I am the more likely I am to report with rose tinted glasses….

Response 8

Yes, this is also possible. However, this would not constitute random measurement error – this would be a systematic measurement error (i.e., people systematically reporting their weight differently depending on their depressive symptoms level, thus creating the appearance of an association when in fact none exists). In this particular section, we are discussing reasons why we might have found no association between weight and subsequent depressive symptoms. One explanation could be that there simply isn’t an association (supporting the notion that depressive symptoms predict weight but weight does not predict depressive symptoms, indicating a potential direction of effect). The alternative explanation is that this apparent null-association might simply be caused by random measurement error in weight. We have included this here to caution the reader that we cannot firmly conclude that weight does not predict depressive symptoms. 

That said, we do agree with the reviewer that systematic measurement error in self-reports could create spurious associations. We have added the following to the discussion:

“Self-reported weight could also involve systematic measurement error (e.g., people with high/low depressive symptom levels may systematically over- or under-report their weight). This could create a spurious association between depressive symptoms and weight.” (p. 26, lines 562-565)

Reviewer #2

Dear authors,

This was an interesting paper looking at not only the relationship of mental health and weight across people, but also at mental health as predictor for weight within people. The only significant relationship found was depression variance as predictor for weight, and the moderating effect of BMI on this relationship.

I think, this study adds to the current literature, but would recommend minor changes:

Thank you very much for your detailed, thorough review of our manuscript and for your constructive comments. We appreciate you taking the time for this. 

Comment 1

Abstract:

Results: Some of the phrasing is harder to follow without the information in the main text and therefore makes it hard to understand the abstract in itself, e.g., what do you mean by ‘within-individual variation’ and ‘dose-dependent’ (I wouldn’t call BMI categories ‘doses’). While it is accurate, for the abstract it might be helpful to describe it in easier terms which are more commonly used.

Response 1

We have now added a brief explanation to the abstract to help the reader understand what we mean by within and between-individual variation: 

“Mental health variables were split into “between-individual” measurements (=the participant’s median score across all timepoints) and “within-individual” measurements (at each timepoint, the difference between the participant’s current score and the median).” (p. 2, lines 38-41)

Additionally, we have removed “dose-dependent” and instead described what we mean by this: 

“We found evidence of a moderation effect of baseline BMI on the association between within-individual fluctuation in depressive symptoms and subsequent weight: The association was only apparent in those with overweight/obesity, and it was stronger in those with obesity than those with overweight (BMI<25kg/m2: 0.011kg per unit of depressive symptom severity [95% CI -0.017 to 0.039]; BMI 25-29.9kg/m2: 0.052kg per unit of depressive symptom severity [95%CI 0.010-0.094kg]; BMI≥30kg/m2: 0.071kg per unit of depressive symptom severity [95%CI 0.013-0.129kg]).” (p. 2, lines 44-51)

Comment 2

CI intervals are sometimes written in square brackets and sometimes not. Should be uniform.

Response 2

The CIs are only in square brackets in the abstract; in this case we have placed them in square brackets because they are nested within round brackets. In the remainder of the manuscript we have enclosed CIs in round brackets.

Comment 3

You mention the reverse direction of association in the results part. It was not mentioned before that you are interested in it and therefore, comes a bit surprisingly.

Response 3

This is mentioned in our aims; we have edited this to make it clearer why we included this:

“We aimed to investigate:

• whether within-individual and between-individual differences in weight are associated with subsequent mental health (to investigate the reverse direction of association and help us better understand the direction of the association)1 (p. 7, lines 31-33)

There is also a footnote with this aim that explains this further: 

“ This objective and the accompanying analyses were not included in the original Statistical Analysis Plan. However, on reviewing our findings regarding the effect of depressive symptoms on subsequent weight, and the moderating effect of baseline BMI, we deemed it important to explore the possibility of a reverse direction of association and assess whether weight predicts subsequent depressive symptoms.” (p. 7)

It is also mentioned in the methods section, in the Analysis section: 

“We used analogous random intercept models to assess the relationship of between-individual and within-individual variation in weight with subsequent mental health domains to explore the reverse direction of association.” (p. 12, lines 253-256)

Comment 4

Background:

l. 92: This sentence could be clearer.

Response 4

We have edited this section to make our point clearer:

“The lack of research on within-individual variation in mental health and weight is partly due to the paucity of datasets with frequent, repeated measurement data, but also because examination of change depends on the extent to which a given variable varies within an individual. When variation is low, it is difficult to assess how changes in one variable influence changes in another.” (p. 5, lines 97-101)

Comment 5

l. 94-100: You mention that COVID gives a unique opportunity to assess the changes, as these variations in mental health are stronger than usual. But could not COVID influence weight via other pathways, e.g., increased sedentary behaviour? You took restrictions into account, but the effect were sometimes extending these restriction periods and would therefore not be covered by the covariate of restrictions.

Response 5

Yes, we agree that it is possible that COVID restrictions may have influenced physical activity and this may have influenced mental wellbeing and weight. We have added a section on this to the discussion:

“Physical activity could also be an important related variable, particularly in light of COVID-19 restrictions. In an observational study during the COVID-19 pandemic, higher BMI was associated with lower reported levels of physical activity (52), and physical activity is linked with depressive symptoms (53). Thus, one alternative explanation of our findings is that changes in physical activity caused both changes in depressive symptoms and weight.” (p. 26, lines 571-576)

Comment 6

l. 107-109 (also related to l.116): How did you define the ‘usual levels’ of mental health? Is that the baseline or mean? Is this difference between their measured level to the usual level the ‘within-individual variation’ you mention in aim i)?

Response 6

Yes, the “usual level” would be the between-individual variation, and the difference between their measured level to the usual level is the “within-individual variation”. We have now edited this sentence to clarify this:

Introduction: 

“Importantly, we explored both how an individual’s general mental health (“between-individual variation”) may affect weight, but also how fluctuations in an individual’s mental health compared to their own usual levels (“within-individual variation”) may influence subsequent weight.” (p. 6, lines 113-117)

In the Methods, in the Analysis section, we then explain how we measured between and within-individual variation:

“We used the median of each person’s monthly scores for the between-individual measurement and calculated the deviation of each monthly score from that person’s median for the within-individual measurement.” (p. 11, lines 235-237)

We have also edited the first section of the discussion to again make this clear:

“We found that within-individual deviations in depressive symptoms from a person’s usual depressive symptom level (with the “usual level” defined as the median depressive symptom level) were positively associated with subsequent weight” (p. 19, line 404 – p. 20, line 407)

We have also made this clearer in the abstract:

“Mental health variables were split into “between-individual” measurements (=the participant’s median score across all timepoints) and “within-individual” measurements (at each timepoint, the difference between the participant’s current score and the median).” (p. 2, lines 38-41)

Comment 7

l.122-123: By ‘these associations’, do you mean all three aims you presented before or just specific ones?

Response 7

Thank you for highlighting this. We agree this was unclear. We only did the interaction analyses for the association of mental health variables with subsequent weight. We have changed the order of the objectives and edited these to make this clearer:

“We aimed to investigate: 

• whether within-individual variation in mental health is associated with weight at the next assessment timepoint (approx. 1 month later) 

• whether between-individual differences in mental health are associated with weight 

• whether the above associations vary by participant characteristics (age, sex, education, occupation, and baseline BMI).

• whether within-individual and between-individual differences in weight are associated with subsequent mental health (to investigate the reverse direction of association and help us better understand the direction of the association)”

(p. 6, line 123 – p. 7, line 133)

Comment 8

Is this study/these analyses exploratory or confirmatory? If they were confirmatory, you should add hypotheses. If they were exploratory, it would be good to mention this briefly.

Response 8

This study was exploratory. We have edited the abstract to clarify this:

“In this exploratory study, individuals with overweight or obesity were more vulnerable to weight gain following higher-than-usual (for that individual) depressive symptoms than individuals with a BMI<25kg/m2.” (p. 2, lines 54-56)

We have also edited the introduction section:

“In the present exploratory study, we use data derived from a sub-study of the population-based and well-phenotyped Fenland study” (p. 5, lines 108-109)

We have also made this clearer in the analysis section: 

“We did not perform adjustments to control Type I error following recommendations for exploratory observational studies” (p. 13, lines 277-278)

We have also edited the discussion to make it clearer that our analyses are exploratory:

“However, the present study was an observational, exploratory analysis and findings should be interpreted with caution.” (p. 21, lines 434-435)

Comment 9

Methods:

As you want to use depressive symptoms as predictive marker for weight gain, it might be useful to look at the relationship between changes in depressive symptoms and weight change.

Response 9

We have opted to use the difference between a person’s score and their median score as an indicator of the fluctuation around a person’s usual level. We have opted to do this as it gives an indication of how an individual’s score varies around their “usual level”. However, we have now also added some supplementary analyses regressing change in mental health symptoms onto change in weight (please see our response to Reviewer 1, Comment 2).

Comment 10

Why did you choose a 1-month gap between mental health and weight measure?

Response 10

One month is the amount of time between measurements, as mental health and weight were assessed once a month during the study period. We chose to do monthly measurements due to the timeframe of the mental health questionnaires (they asked participants about the previous 4 weeks). It was decided to do weight measurements monthly to minimise participant burden. We have clarified this further in the study design & procedure section and in the analysis section:

“We chose monthly measurements since the mental health questionnaires enquired about the previous 4 weeks.” (p. 8, lines 167-168)

“For the within-individual variables only, we added measurements of the previous assessment timepoint to the model (rather than of the same assessment timepoint as weight). This means that we assessed the association between within-individual variation in mental health with weight at the next assessment timepoint (i.e., one month later), rather than examining the association with weight at the same timepoint.” (p. 12, lines 248-252)

Comment 11

As you use baseline BMI and age from the main Fenland study as covariates, it would be of interest when the main Fenland study was conducted or how much time was between the main Fenland study and the COVID app sub-study. This helps to get a feeling how BMI could have changed.

Response 11

Apologies, this was an error in the manuscript; thank you for highlighting this. We did not use BMI from the main Fenland study. We computed baseline BMI by using the first weight measurement the participant provided and the height measurement from the main Fenland study. We have now clarified this:

“We included age (continuous), sex (male/female), ethnicity (White/non-White), education (age when completed full-time education) and occupation (Traditional and modern professional and higher managerial/Lower managerial and intermediate occupations/Technical, semi-routine and routine occupations; categories are based on (34)) from the main Fenland study as covariates. We used height (cm) from the main Fenland study (measured in the clinic by trained staff) to compute baseline body-mass-index (BMI) using the first weight measurement provided by each participant.” (p. 10, lines 210-217)

Comment 12

l.198-199: In a later table you mention that the occupation levels were taken from literature. It might be useful to include this information also in the confounder section.

Response 12

Thank you for the suggestion; we have added this to the confounder section: 

“… and occupation (Traditional and modern professional and higher managerial/Lower managerial and intermediate occupations/Technical, semi-routine and routine occupations; categories are based on [34])” (p. 10, line 214)

Comment 13

l. 228-230: You label the within-individual measures of mental health ‘lagged’, but from the description it is the measure before all the other measures, not after. ‘Lagged’ has more the meaning of ‘later’, so I find it confusing. Please also clarify if all variables (such as timepoint, between-individual measurements, season and weight) are from one timepoint and only lagged within-individual measurements from the timepoint before.

Response 13

To avoid any confusion, we have removed the term “lagged” and explain what we did; we hope this is now clearer:

“Time (measurement time point), between-individual measurements and within-individual measurements of mental health variables, demographics (age, sex, education, occupation), baseline BMI, restriction level and season (month as dummy variables) were included as fixed effects. For the within-individual variables only, we added measurements of the previous assessment timepoint to the model (rather than of the same assessment timepoint as weight). This means that we assessed the association between within-individual variation in mental health with weight at the next assessment timepoint (i.e., one month later), rather than examining the association with weight at the same timepoint.” (p. 11, line 243 – p. 12, line 252)

Comment 14

l.240-241: What would be the sub-groups?

Response 14

Thank you for highlighting this. The term “sub-groups” was unclear here. We have rephrased this to:

“Interaction analyses

To explore potential interactions between mental health domains and baseline characteristics on weight, we included interaction terms of within-individual and between-individual measurements by, respectively, age, sex, ethnicity, education and baseline BMI in the main models.” (p. 13, lines 268 – 272)

The subgroups for BMI are then detailed in lines 275-277 (p. 13):

“To follow up interactions with baseline BMI, we re-fitted the models separately within BMI sub-groups to facilitate interpretation (<25kg/m2, 25-29.9kg/m2, ≥30kg/m2).

We did not detail the subgroups for the other variables since we did not find interaction effects for these and therefore did not do subgroup analyses.

Comment 15

l.239: Clarify which analysis you used. Was another random intercepts regression model?

Response 15

We have specified this by saying:

“To explore potential interactions between mental health domains and baseline characteristics on weight, we included interaction terms of within-individual and between-individual measurements by, respectively, age, sex, ethnicity, education and baseline BMI in the main models.” (p. 13, line 272)

The main models are described in the section above (titled “Main models (mental health & weight)”). This means we took the same models as detailed for the main models, and then added the interaction terms. 

Comment 16

l.245-246: Could be clearer.

Response 16

We have rephrased this a bit; hopefully it is now clear what we mean. Essentially, since interaction terms themselves (e.g., baseline depression * baseline BMI) are very difficult to interpret, we followed up any ‘significant’ interaction effects by refitting the models within subgroups so that we could explore how the relationship between mental health and weight differed for different levels of the ‘interacting’ variable. Since we found a significant effect for depressive symptoms * baseline BMI on weight, we then refitted the model for the association of depressive symptoms and weight in the 3 different groups of baseline BMI (<25, 25-29.9, >=30 kg/m2):

“To follow up interactions with baseline BMI, we re-fitted the models separately within BMI sub-groups to facilitate interpretation (<25kg/m2, 25-29.9kg/m2, ≥30kg/m2).” (p. 13, lines 275-277)

Comment 17

You sometimes mention a statistical analysis plan. Did you pre-register the plan somewhere? Then please add a link to the publication.

Response 17

The Statistical Analysis Plan was not pre-registered. It was written up and agreed by the authors of the study prior to commencing the analyses. In the manuscript, we have highlighted (using footnotes) instances where we deviated from the original analysis plan with reasons (p. 7, p. 12, p. 13). We have added a sentence to the Analysis section to clarify:

“Analyses were pre-specified in a statistical analysis plan which was reviewed and agreed by the authors of the study prior to commencing analyses.” (p. 11, lines 228-229)

Comment 18

Results:

While some information is interesting, it might not be needed in the main paper but could be added to the supplemental data, such as the descriptives of people who consented in taking part in the app-study and missing data.

Response 18

We think it is important to provide these descriptive data in the manuscript so that the reader can understand the context. For example, it is important to understand whether those in the analysis dataset differ from those who consented (and whether those with missing data differ from those without missing data) in order to assess risk of selection bias. We would therefore prefer to retain this information in the manuscript. 

Comment 19

l.252-255: It could be clearer that the exclusion of people was from the 2524 who consented to take part, not the 2277 who downloaded the app.

Response 19

We agree this wasn’t phrased very clearly. It previously said we excluded participants with biologically implausible BMI or weight loss. This is not accurate. We excluded individual values of weight that indicated biologically implausible BMI or weight loss, but we did not exclude the entire participant. This was part of the standard data cleaning process. It did not affect the total sample size so we have opted to remove this from this section to avoid any confusion. Thus, there were 2277 participants who downloaded the app, and 144 provided no data at all, resulting in an analysis dataset of 2133 participants. We have rephrased the section to make this clearer:

“Of the 11,469 Fenland participants that were approached, 4031 participants aged 44-70 years consented to take part in the Fenland COVID-19 study and 2524 of these also consented to participate in the app sub-study. Of these, 2277 participants downloaded the app. We excluded participants who reported neither mental health nor weight data (n=144). Our analysis dataset included 2133 participants.” (p. 13, lines 281 – p. 14, line 287). 

Comment 20

Variables in tables should all be uniformly written, e.g., in S1 variables are written in all capital except anxiety. Overall, style, font size, and font should be consistent across all tables and figures. Are page numbers needed in supplementals?

Response 20

We have reviewed all tables and ensured they have uniform formatting and style. The journal requires us to upload each supplementary table as a separate file; we have therefore not added page numbers. 

Comment 21

The figures could be more ‘professional’, e.g., figure 1 the label for the legend is ‘bmi_cat’ which could be nicer with ‘BMI cat’ or even ‘BMI categories’ if space permits. Even abbreviation is used, it should be explained. For figure 2, the labels ‘within-ind. Variation…’ should look the same as the ‘weight’ label.

Response 21

We have updated the legend titles for the graphs in Figure 1 to “BMI category”. Additionally, we improved the resolution of the graphs.

For Figure 2, please see our response to your next comment. 

Comment 22

Figure 2: What does the black line with triangles show? The distribution of the weight on the axis could be more equal. I am aware that people are in different BMI categories and therefore the weight is different, but to be able to compare changes, the distance might be good to keep equal, e.g., 0.5kg

Response 22

On further reflection, we feel that Figure 2 does not add much to the manuscript and is not particularly helpful or descriptive. In the interest of keeping the manuscript concise (and since we have added new sections in response to review comments), we have opted to remove this figure. 

Comment 23

l.290: The supplemental data here are S5 and 6, but afterwards you have the data S1-4. It might be more logical to have the supplemental data in order of appearance in the text.

Response 23

We have updated the Table numbering so that it is in order of appearance. 

Comment 24

S6: The label at the separate document and the pdf manuscript are not the same. Usually, you explain that values in bold are CI who do not include 0; this is missing here.

Response 24

We have updated the Table caption in the manuscript to match the separate file. We have also added “Estimates of odds ratios with confidence intervals not including 1 are marked in bold.”

Comment 25

l 291-292: You explain how many of the 2277 who downloaded the app did not provide data. Would you not exclude these participants as you would need mental health data for your analyses? Then the information would be more appropriate in the participants section and the final number of participant data would be reduced. If you still included it as participants gave weight and other data, it might be more useful to know how many of the 2133 participants which you included in your analyses did provide no mental health data at all.

Response 25

We agree that it was confusing to mention this in the Missing data section and we have therefore removed “Of the 2277 participants who downloaded the app, 144 participants provided no data at all” from this section. We have moved this to the Participants section, and it should now be clearer how we arrived at our analysis sample (see our response to Comment 19).

Comment 26

l. 301: ‘weight value of 0.045kg’, ‘of’ missing

Response 26

We have edited the sentence. 

Discussion:

Comment 27

l. 389-391: You say here that your research is in line with previous findings from cross-sectional studies that depression is related to weight. However, when you analyse the association of between-individual depression and weight, it is almost like a cross-sectional study. But you didn’t find a relationship in the between-individual differences and weight. Therefore, you do not support the results of previous cross-sectional studies. Could you explain why you did not find a relationship. Also in context of finding an association of within-individual variation in depression.

Response 27

Thank you for pointing this out. We had meant that our findings support the general idea that depressive symptoms are somehow related with weight. But you are correct – technically our results on between-individual variation in depressive symptoms are at odds with cross-sectional research. We have now stated this more clearly and have put forward one possible reason, but we also acknowledge that there may be other reasons. The section now reads:

“Our findings are consistent with the general notion that depressive symptoms are associated with weight. Previous cross-sectional and prospective research has indicated that higher levels of depression are associated with higher weight, BMI and weight-related outcomes such as food intake and reduced physical activity (6,12–16). For example, in a longitudinal study, depression predicted higher 7-year increase in BMI (std. β = 0.025) and waist circumference (std. β = 0.028) (16). Our study adds to these findings by exploring repeated short-term measurements rather than extrapolating over longer timeframes, and by elucidating how fluctuations within individuals predict weight one month later. Interestingly, some previous studies have found associations of between-individual variation in depressive symptoms with weight (e.g. 41,42), whereas we found none. One key difference is that the majority of previous research has involved dichotomising depression (i.e., clinical depression vs. no clinical depression), whereas we used depressive symptoms as a continuous outcome. However, evidence on the association of depression with weight is typically mixed (1) and there may be a multitude of reasons for variations in results.” (p. 21, line 445 – p. 22, line 462). 

Comment 28

l. 420-431: You explain in this paragraph that previous literature showed that stress and anxiety were related to higher weight. As you already tried to explain your findings in the previous section (which are in contrast with the literature highlighted in this paragraph), it is not clear how the results of this paragraph are related to your study. You mention that longitudinal studies such as yours are missing, but I miss the link to the previous section and your results. Why do you think more longitudinal studies are needed when your findings contradict previous studies?

Response 28

The previous section stated that we found no association of stress/anxiety with weight, and we name some potential reasons, and we cite some research that confirms our finding. The purpose of this next section is to acknowledge that, although we found no association and although some previous research is in line with that, there is also research that contradicts our finding – i.e., there are also studies (including systematic reviews) that do show an association between stress/anxiety and weight. We then go on to show that the reviews on this topic only include limited longitudinal studies, which might explain the differences to our findings. We have rephrased this section and cut down the words; hopefully it is now clearer:

“However, it should be noted that, in contrast to our findings, there is also evidence from systematic reviews indicating both stress and anxiety can be associated with higher weight. (2–4). These reviews primarily draw on cross-sectional studies. This may explain why our findings differ.” (p. 23, lines 482-487)

Reviewer #3

This is an excellent study that is extremely welcome within the literature – it reflects a valuable longitudinal dataset which is a rarity. The question asked is extremely topical. It is a thorough manuscript that is written with clarity. I thought the way that unit change in depression impacted weight was really well explained. I just have four points that the authors might consider.

Thank you for taking the time to review our manuscript, and for your constructive and encouraging feedback, it is very much appreciated. 

Comment 1

1. The quality of the figures is noticeably quite low (or at least how they have come out in the pdf I have downloaded) – I would highly recommend finding a way to improve the quality.

Response 1

We have recreated the graphs with a higher resolution (300dpi); we hope they now turn out better. Please note the image looks very crips in our file, but we noticed it turns blurry in the PDF generated upon submission. We will work with the copy editors to try and rectify this. 

Comment 2

2. It is good to see that the small effect sizes are reflected on in the discussion section and as I mentioned before the way relationship between the predictor and the outcome is well explained. It would be helpful to have an understanding of the variance of the dependent variables explained by the models (R-squared value? – apologies if it is in there and I missed it).

Response 2

We previously considered including pseudo R-squared values for the models in Table 3. However, we had decided not to report these in the manuscript since we felt that reporting R-squared is not necessarily helpful here. It is a measure of goodness of fit. In this case it would tell us the amount of variance explained by the overall model (exposure + all covariates). However the aim of our models is to model the association of individual exposures with the outcome, adjusting for covariates. Hence we are not really interested in the goodness of fit of the overall model; we are primarily interested in the regression coefficient for each exposure with the outcome. The covariates are included in the model to improve the precision, but not in order to assess their combined predictive power to predict weight. For example, for the models in Table 3, the pseudo R-squared values are quite high, but this is because they include baseline BMI, which explains a large amount of variance. We feel that reporting R-squared would be more relevant when developing a prediction model. Therefore we opted not to include them here. 

Comment 3

3. Did you look at the extent to which across time, rise in depression score led to missing values the following month?

Response 3

We did not model how changes in depressive symptoms over time predict missing values in the following month. We limited our missing data analysis to examining how baseline characteristics predict overall missingness. As part of this, we also examined whether baseline depressive symptoms score was associated with having missing data on any of the exposure/outcome variables and found that those with worse mental health at baseline were more likely to have missing data for weight, and we have discussed potential implications for the generalisability of the findings in the discussion. 

 Comment 4

4. Finally, was this study pre-registered in line with open research practice? This would be an ideal study to pre-register in order to increase rigour as this kind of dataset could be vulnerable to p-hacking…

Response 4

We fully agree that this study would have been ideal to pre-register. Unfortunately this study was not pre-registered. We did, however, agree an analysis plan within the study team prior to commencing analyses, and we have now clarified this in this Analysis section:

“Analyses were pre-specified in a statistical analysis plan which was reviewed and agreed by the authors of the study prior to commencing analyses.”

We have also highlighted throughout the manuscript (using footnotes) where we have deviated from the original plan. Hopefully this helps demonstrate our commitment to rigour and transparency. We have also (at the suggestion of Reviewer 1) emphasised that this study is exploratory and therefore at risk of inflation of type 1 error, and that results should be interpreted with necessary caution. We do agree though that open science processes are important and we will pre-register future observational, exploratory studies wherever possible.

---

## [Decision Letter · Decision Letter 1]

25 Sep 2023

PONE-D-23-00921R1The relationship of within-individual and between-individual variation in mental health with bodyweight: A longitudinal studyPLOS ONE

Dear Dr. Mueller,

Thank you for submitting your manuscript to PLOS ONE. After careful consideration, we feel that it has merit but does not fully meet PLOS ONE’s publication criteria as it currently stands. Therefore, we invite you to submit a revised version of the manuscript that addresses the points raised during the review process. The few comments raised by reviewers are straightforward. Please submit your revised manuscript by Nov 09 2023 11:59PM. If you will need more time than this to complete your revisions, please reply to this message or contact the journal office at plosone@plos.org. Please include the following items when submitting your revised manuscript:A rebuttal letter that responds to each point raised by the academic editor and reviewer(s). You should upload this letter as a separate file labeled 'Response to Reviewers'.A marked-up copy of your manuscript that highlights changes made to the original version. You should upload this as a separate file labeled 'Revised Manuscript with Track Changes'.An unmarked version of your revised paper without tracked changes. You should upload this as a separate file labeled 'Manuscript'.If applicable, we recommend that you deposit your laboratory protocols in protocols.io to enhance the reproducibility of your results. Protocols.io assigns your protocol its own identifier (DOI) so that it can be cited independently in the future. For instructions see: https://journals.plos.org/plosone/s/submission-guidelines#loc-laboratory-protocols. Additionally, PLOS ONE offers an option for publishing peer-reviewed Lab Protocol articles, which describe protocols hosted on protocols.io. Read more information on sharing protocols at https://plos.org/protocols?utm_medium=editorial-email&utm_source=authorletters&utm_campaign=protocols.

We look forward to receiving your revised manuscript.

Kind regards,

Nafis Faizi, MD, MPH

Academic Editor

PLOS ONE

Journal Requirements:

Reviewers' comments:

Reviewer's Responses to Questions

**Comments to the Author**

1. If the authors have adequately addressed your comments raised in a previous round of review and you feel that this manuscript is now acceptable for publication, you may indicate that here to bypass the “Comments to the Author” section, enter your conflict of interest statement in the “Confidential to Editor” section, and submit your "Accept" recommendation.

Reviewer #1: All comments have been addressed

Reviewer #2: (No Response)

2. Is the manuscript technically sound, and do the data support the conclusions?

Reviewer #1: Yes

Reviewer #2: Yes

3. Has the statistical analysis been performed appropriately and rigorously? 

Reviewer #1: Yes

Reviewer #2: Yes

4. Have the authors made all data underlying the findings in their manuscript fully available?

Reviewer #1: No

Reviewer #2: No

5. Is the manuscript presented in an intelligible fashion and written in standard English?

Reviewer #1: Yes

Reviewer #2: Yes

6. Review Comments to the Author

Reviewer #1: I thought the authors did a very comprehensive job of addressing my comments and thank them for being open to some alternative suggestions/ideas.

I suggest one very minor final change. I think the title should be called 'an exploratory longitudinal study' given that the authors describe the study as such in the manuscript and the p value isn't correct for multiple comparisons. Assuming the authors are happy to do this then I don't need/want to see this again, and would prefer the editor signs off on my behalf once the revision comes in.

Well done on this work.

Reviewer #2: Dear authors,

Thank you very much for implementing my and the other reviewers’ suggestions. I think, it reads much clearer now.

There are just very minor suggestions to further improve clarity:

Results:

ll. 169-172: In the methods section, it still sounds like you excluded the whole participant if BMI or weight loss were implausible. Later in the results section (ll. 271-273) and your response to my comment, it sounds more like only the datapoint of BMI or weight loss was removed, while the rest of the participant’s data were still used.

ll. 276-278: You added the average number of values for each mental health outcome, which is a good idea, I think. It would be good to have an indication of how long the assessment period was and therefore, how many datapoints they could maximally enter. In your methods section, you give the month/year when participants can start download the app and enter datapoints to it, and when the last month/year is for data entry. To make it easier for the reader, it would be helpful if you could calculate the number of months and add this information here to the results section. This would help the reader to bring the average number of values into context and to determine if people entered many or only few data points.

Table 1: You have “a” described under your table, but “4” as a footnote. As both describes variables in the table, also the information about SES could be under the table as “b”.

Figure 1: Sorry for being pedantic, for graph b, the label “time” for the x-axis is missing. As you mentioned in your answer to reviewer 1, please do work with the editors to get a higher quality figure. It is really hard to read this figure.

7. PLOS authors have the option to publish the peer review history of their article (what does this mean?). If published, this will include your full peer review and any attached files.

Reviewer #1: **Yes: **Eric Robinson

Reviewer #2: No

---

## [Author Response · Author response to Decision Letter 1]

25 Sep 2023

Reviewer 1

Comment 1

Reviewer #1: I thought the authors did a very comprehensive job of addressing my comments and thank them for being open to some alternative suggestions/ideas.

I suggest one very minor final change. I think the title should be called 'an exploratory longitudinal study' given that the authors describe the study as such in the manuscript and the p value isn't correct for multiple comparisons. Assuming the authors are happy to do this then I don't need/want to see this again, and would prefer the editor signs off on my behalf once the revision comes in.

Well done on this work.

Response 1

Thank you very much, we really appreciate the reviewers taking the time to review and help improve our manuscript. Thank you for the suggestion, we updated the title as suggested:

“The relationship of within-individual and between-individual variation in mental health with bodyweight: An exploratory longitudinal study”

Reviewer 2

Comment 1

Reviewer #2: Dear authors,

Thank you very much for implementing my and the other reviewers’ suggestions. I think, it reads much clearer now.

There are just very minor suggestions to further improve clarity:

Response 1

Thank you very much, we really appreciate you taking the time to review and help improve our manuscript.

Comment 2

Results:

ll. 169-172: In the methods section, it still sounds like you excluded the whole participant if BMI or weight loss were implausible. Later in the results section (ll. 271-273) and your response to my comment, it sounds more like only the datapoint of BMI or weight loss was removed, while the rest of the participant’s data were still used.

Response 2

Thank you for spotting this. We have amended the text to clarify that we excluded only specific data points rather than the entire participant: 

“For the present analysis, we excluded weight entries that indicated a biologically implausible BMI of <12 or >70kg/m2 [26,27] or biologically implausible weight loss between two timepoints as defined by Chen et al. [28].” (p. 8, line 173)

Comment 3

ll. 276-278: You added the average number of values for each mental health outcome, which is a good idea, I think. It would be good to have an indication of how long the assessment period was and therefore, how many datapoints they could maximally enter. In your methods section, you give the month/year when participants can start download the app and enter datapoints to it, and when the last month/year is for data entry. To make it easier for the reader, it would be helpful if you could calculate the number of months and add this information here to the results section. This would help the reader to bring the average number of values into context and to determine if people entered many or only few data points.

Response 3

Thank you for pointing this out. We have now added a sentence to the methods section to clarify:

“Those in the app sub-study were able to download the app from 6th August 2020, and the study finished on 30th April 2021 (when the app also closed to further measurement entry). The number of monthly measurements participants could provide depended on the timepoint when they entered the study (e.g., a participant entering in August 2020 could add a total of 9 measurements up until April 2021, a participant entering in September could add 8, etc.)”. (p. 7, lines 157-160)

Comment 4

Table 1: You have “a” described under your table, but “4” as a footnote. As both describes variables in the table, also the information about SES could be under the table as “b”. 

Response 4

We have now changed it so that it’s consistent (using “a” and “b” to add notes at the bottom of the table to describe variables in more detail). (Table 1, pp. 13-14)

Comment 5

Figure 1: Sorry for being pedantic, for graph b, the label “time” for the x-axis is missing. As you mentioned in your answer to reviewer 1, please do work with the editors to get a higher quality figure. It is really hard to read this figure.

Response 5

Thank you for spotting this. We have edited the figure so that the x-axis labelling is now visible. Also, apologies that the figure is coming out blurry. It is unclear why it is appearing so blurry in the pdf for peer review. It looks very crisp on my screen, and the file has a resolution of 300dpi, which should be sufficient. If it turns up blurry during copy editing, we will make sure to address this.

---

## [Decision Letter · Decision Letter 2]

16 Nov 2023

The relationship of within-individual and between-individual variation in mental health with bodyweight: An exploratory longitudinal study

PONE-D-23-00921R2

Dear Dr. Mueller,

We’re pleased to inform you that your manuscript has been judged scientifically suitable for publication and will be formally accepted for publication once it meets all outstanding technical requirements.

Kind regards,

Giulia Ballarotto

Academic Editor

PLOS ONE

Additional Editor Comments (optional):

Reviewers' comments:

Reviewer's Responses to Questions

**Comments to the Author**

1. If the authors have adequately addressed your comments raised in a previous round of review and you feel that this manuscript is now acceptable for publication, you may indicate that here to bypass the “Comments to the Author” section, enter your conflict of interest statement in the “Confidential to Editor” section, and submit your "Accept" recommendation.

Reviewer #1: All comments have been addressed

Reviewer #2: All comments have been addressed

2. Is the manuscript technically sound, and do the data support the conclusions?

Reviewer #1: Yes

Reviewer #2: Yes

3. Has the statistical analysis been performed appropriately and rigorously? 

Reviewer #1: Yes

Reviewer #2: Yes

4. Have the authors made all data underlying the findings in their manuscript fully available?

Reviewer #1: No

Reviewer #2: No

5. Is the manuscript presented in an intelligible fashion and written in standard English?

Reviewer #1: Yes

Reviewer #2: Yes

6. Review Comments to the Author

Reviewer #1: (No Response)

Reviewer #2: (No Response)

7. PLOS authors have the option to publish the peer review history of their article (what does this mean?). If published, this will include your full peer review and any attached files.

Reviewer #1: **Yes: **Eric Robinson

Reviewer #2: **Yes: **Jennifer Gatzemeier

---

## [Editor Report · Acceptance letter]

30 Nov 2023

PONE-D-23-00921R2 

The relationship of within-individual and between-individual variation in mental health with bodyweight: An exploratory longitudinal study 

Dear Dr. Mueller:

I'm pleased to inform you that your manuscript has been deemed suitable for publication in PLOS ONE. Congratulations! Your manuscript is now with our production department. 

Kind regards, 

on behalf of

Dr Giulia Ballarotto 

Academic Editor

PLOS ONE